# HyperINF: Unleashing the HyperPower of Schulz's Method for Data Influence Estimation

## Abstract

Influence functions provide a principled method to assess the contribution of individual training samples to a specific target. Yet, their high computational costs limit their applications on large-scale models and datasets. Existing methods proposed for influence function approximation have significantly reduced the computational overheads. However, they mostly suffer from inaccurate estimation due to the lack of strong convergence guarantees from the algorithm. The family of *hyperpower methods*[1] are well-known for their rigorous convergence guarantees on matrix inverse approximation, while the matrix multiplication operation can involve intractable memory and computation costs on large-scale models. We propose HyperINF, an efficient and accurate influence function approximation method which leverages the *hyperpower method*, specifically Schulz's iterative algorithm. To deal with the computation-intensive matrix multiplication, we incorporate the *generalized fisher information* (GFIM) as a low-rank approximation of the Hessian matrix, which reduces the memory and computation overheads to constant costs independent of ranks on LoRA-tuned models. We first demonstrate the superior accuracy and stability of HyperINF compared to other baselines through a synthetic convergence simulation for matrix inversion. We further validate the efficacy of HyperINF through extensive real-world data attribution tasks, including mislabeled data detection and data selection for LLM and VLM fine-tuning. On LoRA-tuned models, HyperINF achieves superior downstream performance with minimal memory and computational overhead, while other baselines suffer from significant degradation. Our codebase is available at https://anonymous.4open.science/r/HyperINF-B702.

## 1 Introduction

Large foundation models have demonstrated remarkable capabilities on a great variety of tasks across language, vision and audio modalities (Touvron et al., 2023; Liu et al., 2023a; OpenAI et al., 2024; Bai et al., 2023). Recently, extensive data-centric studies illustrate that training data plays an essential role in the model's downstream performance (Hoffmann et al., 2022; Gao et al., 2020; Penedo et al., 2023; Wang et al., 2018; Gunasekar et al., 2023; Lee et al., 2023; Longpre et al., 2023b). Therefore, the community calls for an efficient and effective data attribution method which identifies the most beneficial training samples without introducing large computation overheads on large-scale models and data pools. As one of the most principled data attribution methods, influence function quantifies the impact of each training sample on model's prediction on a validation set (Hampel, 1974; Koh & Liang, 2020). Despite the efficacy of influence function and its variants (Kwon et al., 2024; Koh & Liang, 2020; Pruthi et al., 2020; Guo et al., 2021; Wang et al., 2019b; Kong et al., 2021), the Hessian inverse operation involved in the formulation introduces intractable memory and computation costs, which hinders its wide application on large models.

To mitigate the computation overheads, a series of methods are proposed to estimate the values of influence function with lower costs. Agarwal et al. (2017) proposed LiSSA, which iteratively estimates the value of the Hessian-vector product. However, the convergence of the algorithm is not guaranteed, which could largely diverge from the correct value after several iterations. Recently, Kwon et al. (2024) introduced DataInf as a closed-form approximation of the Hessian matrix,

---

[1] A hyperpower method is defined as a function $\Phi(A, X)$ on matrices $A$ and $X$, where $A^{-1}$ is the targeted matrix inverse (Petković, 1995).

Table 1: Complexity Comparison between Exact (Gaussian Elimination), LiSSA, DataInf and HyperINF. Computational and memory complexities are obtained on a LoRA-tuned model with dimension $d \in \mathbb{N}$ and rank $r \in \mathbb{N}$. Assume the dimension of the LoRA matrices is identical across $L$ different layers.

| Complexity | Exact (Gaussian Elimination) | LiSSA | DataInf | HyperINF w. GFIM | HyperINF w. FIM |
|---|---|---|---|---|---|
| $H^{-1}$ Computation | $O(r^2 d^2 L + r^3 d^3 L)$ | - | $O(rdL)$ | $O(d^3 L)$ | $O(r^3 d^3 L)$ |
| $H^{-1} g$ Computation | $O(r^2 d^2 L + r^3 d^3 L)$ | $O(r^2 d^2 L)$ | $O(rdL + r^2 d^2 L)$ | $O(d^3 L + rd^2 L)$ | $O(r^3 d^3 L + r^2 d^2 L)$ |
| Memory | $O(r^2 d^2)$ | $O(r^2 d^2)$ | $O(rd)$ | $O(d^2)$ | $O(r^2 d^2)$ |

which further reduces the complexity. However, the error bound of the method is quadratic to the scale of the matrix Kwon et al. (2024), which is vulnerable to downstream performance degradation.

To further improve the accuracy of hessian-inverse estimation, the hyperpower method is considered a promising alternative with rigorous convergence guarantees (Garnett et al., 1971; Behera et al., 2024). However, the hyperpower method iteratively applies matrix multiplication operation, which introduces intractable memory and computation costs, especially on large-scale networks. To improve the influence function estimation accuracy within tractable computations, we thereby introduce HYPERINF as a novel approximation method by incorporating the hyperpower method, specifically Schulz's iterative algorithm (Petković, 1995). To address the costs from matrix multiplication, we use the generalized fisher information matrix (GFIM) (Hu & Li, 2024) as a low-rank approximation of the Hessian matrix, with a theoretical proof. Specifically, on LoRA-tuned models, the memory and computational costs are reduced to a constant value which is independent of the LoRA ranks. We show that HYPERINF with GFIM demonstrates superior accuracy benefit from rigorous convergence guarantee while incurring low computational overheads compared to other baseline methods. From extensive experiments on LLM and VLM, HYPERINF can effectively identify the most helpful and mislabelled data points, which improves the data attribution interpretability and finetuning efficiency.

**Our Contributions.** We summarize our main contributions as follows:

- We leverage the generalized fisher information matrix (GFIM) to derive a novel low-rank formulation of influence function Equation 5, which largely improve the efficiency of influence function computations on large-scale models;

- We demonstrate that the Schulz's method (Equation 7) significantly improves stability and accuracy of the approximation of hessian inversion, which further yields more accurate influence scores for large-scale data attribution;

- We propose HYPERINF as an accurate and efficient influence functions approximation method by applying GFIM and the Schulz's method. We further verify the empirical efficiency and effectiveness of HYPERINF across a range of extensive experiments, including mislabeled data detection (§ 4), data selection for LLM fine-tuning (§ 5.2), and instruct-tuning data selection for VLM pretraining (§ 5.3).

## 2 PRELIMINARIES

We first revisit the influence function formulation with two existing approximation methods LiSSA and DATAINF.

**Setup.** The data attribution problem aims to assess each data point in the training set $\mathcal{D}^{\text{train}} = \{(\boldsymbol{x}_i, y_i)\}_{i=1}^n$ according to their impact to the model's performance on a targeted validation set $\mathcal{D}^{\text{val}} = \{(\boldsymbol{x}_i^{\text{val}}, y_i^{\text{val}})\}_{i=1}^m$. Given a model $f$ parameterized by $\boldsymbol{\theta}$, the loss function on the $i^{th}$ sample $\{(\boldsymbol{x}_i, y_i)\}$ is denoted as $\ell(y_i, f_{\boldsymbol{\theta}}(\boldsymbol{x}_i))$. We assume the loss function is differentiable and strongly convex, the gradient on the $i^{th}$ sample can be represented as $\nabla_{\boldsymbol{\theta}} \ell_i := \nabla_{\boldsymbol{\theta}} \ell(y_i, f_{\boldsymbol{\theta}}(\boldsymbol{x}_i))$ with respect to $\boldsymbol{\theta}$. The empirical risk minimizer on the entire training set is denoted as $\boldsymbol{\theta}^\star = \arg\min_{\boldsymbol{\theta} \in \Theta} \frac{1}{n} \sum_{i=1}^n \ell(y_i, f_{\boldsymbol{\theta}}(\boldsymbol{x}_i))$.

**Influence Functions.** The influence function quantifies how fast the model parameters would change corresponding to the up-weight of a specific data point. Following Koh & Liang (2020),

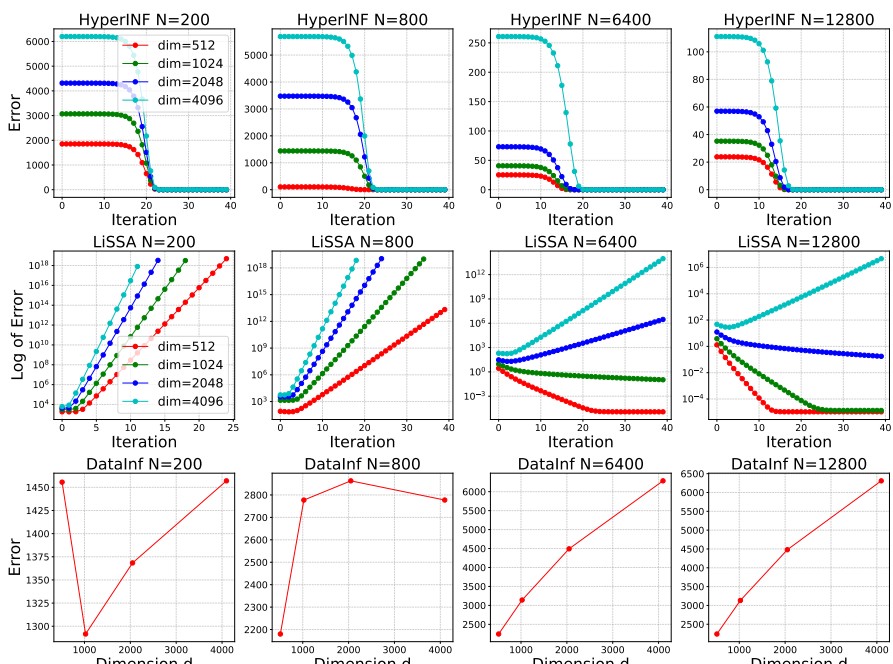

Figure 1: **Convergence test of HYPERINF, LISSA and DATAINF.** We construct $M = \frac{1}{N}\sum_{i=1}^{N} s_i s_i^\top + \lambda I$ and apply various methods to approximate the inverse hessian-vector product $M^{-1}v$, where $s_i \in \mathbb{R}^d$, $v \in \mathbb{R}^d$ are randomly generated from standard normal distribution. Only HYPERINF can converge to a low error rate with increasing matrix dimension and sample size while the approximation error from LISSA and DATAINF significantly diverge from the target values. For LISSA, it does converge but only in limited circumstances (e.g. when $N$ is large). We include the results with other distributions in Appendix H.

given an infinitesimally small $\epsilon > 0$, we upweigh the contribution of the $k^{th}$ datapoint $(x_k, y_k)$ by increasing its portion in the loss function: $\theta^{(k)}(\epsilon) := \arg\min_{\theta \in \Theta} \frac{1}{n}\sum_{i=1}^{n} \ell(y_i, f_\theta(x_i)) + \epsilon \ell(y_k, f_\theta(x_k))$. Assume the loss function $\ell(y, f_\theta(x))$ is twice-differentiable and strongly convex in $\theta$, the influence of the $k^{th}$ data sample $(x_k, y_k) \in \mathcal{D}^{\text{train}}$ on $\theta^\star$ is defined as the derivative of $\theta^{(k)}(\epsilon)$ at $\varepsilon = 0$:

$$\mathcal{I}_{\theta^\star}(x_k, y_k) := \frac{d\theta^{(k)}}{d\varepsilon}\bigg|_{\varepsilon=0} = -H(\theta^\star)^{-1}\nabla_\theta \ell_k \tag{1}$$

where $H(\theta) := \nabla_\theta^2\left(\frac{1}{n}\sum_{i=1}^{n} \ell(y_i, f_\theta(x_i))\right)$ is the Hessian matrix of the empirical loss computed on the flattened gradient vectors (Koh & Liang, 2020; Kwon et al., 2024).

We further score the contribution from each training sample according to model's performance on the validation set $\mathcal{D}^{\text{val}}$. For simplicity, we define $\mathcal{I}(x_k, y_k) := -v^\top H(\theta^\star)^{-1}\nabla_\theta \ell_k$ as the influence from the $k^{th}$ datapoint $(x_k, y_k) \in \mathcal{D}^{\text{train}}$ on $\mathcal{D}^{\text{val}}$, where $v = \frac{1}{m}\sum_{i=1}^{m} \nabla_\theta \ell(y_i^{\text{val}}, f_\theta(x_i^{\text{val}}))|_{\theta=\theta^\star}$, representing the gradient on the validation set, the datapoints assigned with *largest negative values*[2] of influence function would lead to the sharpest drop of validation losses, which contribute the most to the training process. In contrast, the datapoints with *largest positive values* could be the toxic samples which sabotage the model training.

**LISSA.** Agarwal et al. (2017) proposed an iterative method to compute the inverse Hessian vector product $H(\theta^\star)^{-1}v$. For $v_0 = v$, LISSA recursively computes the following iteration: $v_j = v + (I - H(\theta^\star))v_{j-1}$. Agarwal et al. (2017) proved that $v_j$ converges to $H(\theta^\star)^{-1}v$ as $j$ increases, when $H(\theta^\star) \preceq I$. In practice, it is often assumed that LISSA converges to $H(\theta^\star)^{-1}v$ after several reasonable numbers of iterations, and applies the approximation $v_j \approx H(\theta^\star)^{-1}v$ to compute the influence function $\mathcal{I}(x_k, y_k) = -v_j^\top \nabla_\theta \ell_k$. However, some works have shown that the stability and convergence from the iterative update are questionable (Basu et al., 2021; Ko et al., 2024).

---

[2]We refer *largest negative values* here as *negative scores with the largest absolute value*.

**DATAINF.** Kwon et al. (2024) proposed a closed-form approximation of the Hessian inverse, which greatly improves the computation efficiency. Firstly, following George et al. (2021), when applying the negative log-likelihood loss function $\ell(y, f_{\boldsymbol{\theta}}(x)) = -\log p(y|f_{\boldsymbol{\theta}}(\boldsymbol{x}))$, the second-order Hessian is equivalent to the Fisher Information Matrix (FIM) *in expectation* (Bartlett, 1953), which only involves first-order computations. Consequently, Kwon et al. (2024) approximate the Hessian inverse leveraging the Sherman-Morrison formula [3]:

$$H(\boldsymbol{\theta})^{-1} \approx \frac{1}{n\lambda} \sum_{i=1}^{n} \left( I_d - \frac{\nabla_{\boldsymbol{\theta}} \ell_i \nabla_{\boldsymbol{\theta}} \ell_i^{\top}}{\lambda + \nabla_{\boldsymbol{\theta}} \ell_i^{\top} \nabla_{\boldsymbol{\theta}} \ell_i} \right) \tag{2}$$

where $G(\boldsymbol{\theta}) := \frac{1}{n} \sum_{i=1}^{n} \nabla_{\boldsymbol{\theta}} \ell_i \nabla_{\boldsymbol{\theta}} \ell_i^{\top}$ stands for the Fisher Information Matrix (FIM). While the computation complexity of Equation 24 is reduced to $\mathcal{O}(d)$, in compromise, the reverse-order operation Equation 23 incurs a $\mathcal{O}(d^2)$ error (Kwon et al., 2024). When applying to large-scale models, it could risk a large approximation error.

# 3 HYPERINF: EFFICIENT AND ACCURATE DATA INFLUENCE APPROXIMATION VIA THE HYPERPOWER METHOD

We introduce HYPERINF as an accurate yet efficient approximation method for influence function, which leverages generalized Fisher Information Matrix (GFIM) proposed by Yang et al. (2022) and Hu & Li (2024), and Schulz's hyperpower method (Petković, 1995). We begin by providing a theoretical proof of Hessian matrix approximation for large models using GFIM, followed by a demonstration of Schulz's iteration for approximation of the matrix inverse.

## 3.1 LARGE-SCALE HESSIAN APPROXIMATION USING GENERALIZED FISHER INFORMATION

The second-order gradients often incur intensive computations and instability on large-scale networks. Therefore, we conduct several approximations on Hessian matrix when applying Equation 1 on LoRA-tuned models.

**Block-wise Diagonal Approximation.** In deep transformer-structured networks, the Hessian matrix is observed to be approximately block-wise diagonal according to (Zhang et al., 2024a;b). We, therefore, apply a *block-wise diagonal approximation* on the Hessian inverse in Equation 1. Given a neural network as a compositional function $f_{\boldsymbol{\theta}}(x) = f_{\boldsymbol{\theta}_L} \circ \cdots \circ f_{\boldsymbol{\theta}_1}(x)$ where for $l \in [L]$, we compute the hessian inverse on each parameter block which yields a sparse estimation as $\mathrm{diag}(H_1(\boldsymbol{\theta})^{-1}, \ldots, H_L(\boldsymbol{\theta})^{-1})$ (Grosse et al., 2023b).

**Connection between Generalized Fisher Information and Hessian Matrix.** Suppose that we train the model to minimize the negative log-likelihood objective: $\ell(y, f_{\boldsymbol{\theta}}(x)) = -\log p(y \mid f_{\boldsymbol{\theta}}(x))$ for all $(x, y) \in \mathcal{X} \times \mathcal{Y}$, where $p(\cdot)$ is the probability density function and $\mathcal{X}, \mathcal{Y}$ are input and output space, respectively. According to Bartlett's second identity (Bartlett, 1953), the second momentum of first-order gradient (i.e. Fisher Information Matrix) is equivalent to the second-order gradient matrix (Hessian) in expectation:

$$\mathbb{E}_{X,Y \sim p(X), p(Y|f_{\theta}(X))} \left[ \nabla_{\boldsymbol{\theta}}^2 \ell(Y, f_{\boldsymbol{\theta}}(X)) \right] \tag{3}$$
$$= \mathbb{E}_{X,Y \sim p(X), p(Y|f_{\theta}(X))} \left[ \nabla_{\boldsymbol{\theta}} \ell(Y, f_{\boldsymbol{\theta}}(X)) \left( \nabla_{\boldsymbol{\theta}} \ell(Y, f_{\boldsymbol{\theta}}(X)) \right)^{\top} \right].$$

Since Equation 3 replaces the second-order gradient with stable and tractable first-order gradients, the Fisher Information Matrix (FIM) is widely adopted as a valid approximation of Hessian matrix in deep networks (Grosse et al., 2023a; Kwon et al., 2024; Barshan et al., 2020). We further extend the Generalized Fisher Information Matrix (GFIM) (Hu & Li, 2024) to yield a low-rank formulation of influence function. With some idealized assumptions, we claim the Lemma 3.1 following the insights from Yang et al. (2022) and Hu & Li (2024).

**Lemma 3.1.** *Given the matrix-form gradient on a parameter block $\boldsymbol{\theta}$ as $\boldsymbol{g} = \boldsymbol{g}(\boldsymbol{\theta}; x, y) \in \mathbb{R}^{d \times r}$, which can be flattened to a vector by $\mathrm{vec}(\boldsymbol{g}) \in \mathbb{R}^{1 \times rd}$. Let $\otimes$ denotes the Kronecker product, $I_r$ denotes $r \times r$ identity matrix. Assume that each column of the sample gradient $\boldsymbol{g} = \boldsymbol{g}(\boldsymbol{\theta}; x, y) \in \mathbb{R}^{d \times r}$ is independent and identically distributed random vector with zero mean under the distribution $p(y \mid x, \boldsymbol{\theta})$ for any $\boldsymbol{\theta}$. We have:*

$$\mathbb{E}\left[\mathrm{vec}(\boldsymbol{g})\,\mathrm{vec}(\boldsymbol{g})^{\top}\right] = \mathbb{E}\left[I_r \otimes \left(\frac{1}{r} \boldsymbol{g} \boldsymbol{g}^{\top}\right)\right].$$

---

[3]For simplicity, we denote $\ell_i := \ell(y_i, f_{\boldsymbol{\theta}}(\boldsymbol{x}_i))$

*In addition (Equation 3), it holds:*

$$\mathbb{E}\left[I_r \otimes \frac{1}{r}\boldsymbol{g}\boldsymbol{g}^\top\right] = \mathbb{E}[H(\text{vec}(\boldsymbol{\theta}))].$$

Following Lemma Theorem 3.1, we further estimate a hessian-gradient product using GFIM, corresponding to the $(H(\boldsymbol{\theta}^\star)^{-1}\nabla_{\boldsymbol{\theta}}\ell_k)$ term in Equation 1. Given an invertible matrix $A$, we have $(I_r \otimes A)^{-1} = I_r \otimes A^{-1}$. Therefore, denote the GFIM matrix as $G(\boldsymbol{\theta}) \triangleq (\boldsymbol{g}\boldsymbol{g}^\top) \in \mathbb{R}^{d \times d}$ for any matrix $\boldsymbol{v} \in \mathbb{R}^{d \times r}$, it holds that:

$$H(\text{vec}(\boldsymbol{\theta}))^{-1}\text{vec}(\boldsymbol{v}) \approx \left[I_r \otimes (\frac{1}{r}\boldsymbol{g}\boldsymbol{g}^\top)^{-1}\right]\text{vec}(\boldsymbol{v}) = \text{vec}(G(\boldsymbol{\theta})^{-1}\boldsymbol{v}). \tag{4}$$

Consider a LoRA-tuned model with LoRA dimension $d$ and rank $r$. We assume that each column in one LoRA block $\Delta W \in \mathbb{R}^{d \times r}$, corresponding to each rank, is i.i.d. distributed with zero mean. In the ideal case that the model is trained to converge with $\mathbb{E}(-\nabla_{\boldsymbol{\theta}}\log p(y|x,\boldsymbol{\theta})) = 0$, the zero-mean assumption on the columns of gradient matrices could stand. Thus, we apply Equation 4 to approximate the original Hessian-gradient product. To further guarantee that $G(\boldsymbol{\theta})$ is invertible, we add a damping factor $\lambda I_d$ to the GFIM matrix following Martens (2010).

We eliminate the constant in Equation 4 then derive the final formula of HYPERINF influence score. On a specific datapoint $\{\boldsymbol{x}_k, y_k\} \in \mathcal{D}^{\text{train}}$, denote the *unflattened* gradient on a parameter block $\boldsymbol{\theta}$ as $\boldsymbol{g}_k(\boldsymbol{\theta})$, we compute:

$$\mathcal{I}_{\text{HYPERINF}}(\boldsymbol{x}_k, y_k) := -\boldsymbol{g}_v^\top(G(\boldsymbol{\theta}^\star) + \lambda I_d)^{-1}\boldsymbol{g}_k(\boldsymbol{\theta}), \tag{5}$$

where $\boldsymbol{g}_v = \frac{1}{m}\sum_{i=1}^{m}\nabla_{\boldsymbol{\theta}}\ell(y_i^{\text{val}}, f_{\boldsymbol{\theta}}(\boldsymbol{x}_i^{\text{val}}))|_{\boldsymbol{\theta}=\boldsymbol{\theta}^\star} \in \mathbb{R}^{d \times r}$, representing the average *unflattened* gradient on $\boldsymbol{\theta}$ on the validation set.

### 3.2 MATRIX INVERSE APPROXIMATION WITH SCHULZ'S METHOD

**Schulz's method (Petković, 1995).** To compute the inverse of one matrix $A$, the hyperpower iterative family of matrix iteration methods has attracted the attention of many researchers due to its rigorous convergence guarantee (Altman, 1960; Garnett III et al., 1971; Bazán & Boos, 2018):

$$X_{t+1} = X_t(I + T_t + T_t^2 + \dots + T_t^{p-1}), \quad T_t = I - AX_t \tag{6}$$

The iterative approach requires $p$ matrix-matrix multiplications per iteration and has an order of convergence $p$ (Bazán & Boos, 2018). When choosing $p = 2$, it yields the Schulz iteration, which can also regarded as a by-product of the Newton method applied to the non-linear equation $f(X) = A - X^{-1}$:

$$X_{t+1} = X_t + X_tY_t, \quad Y_t = I - AX_t \tag{7}$$

It is proved by Ben-Israel & Cohen (1966) and Petković (1995) that with a proper initialization, Schulz's method would converge to $A^{-1}$ in the order of convergence at least $p = 2$. We provide the complete proof of convergence in Appendix C. Compared to other conventional matrix inverse algorithms (e.g. gaussian elimination, conjugate gradient, GMRES), Schulz's method demonstrates superior accuracy in terms of error rate and significant efficiency gains from the GPU acceleration on matrix multiplications. We include more details in Appendix G. With the convergence test on matrix inversion (section 4), we show that starting from a small identity matrix or random gaussian initialization, Equation 7 could converge to a desirable error rate in finite steps ($t¡20$). We provide the pseudo-code in Algorithm 1.

**Summary.** We hereby provide the holistic view of the HYPERINF algorithm for influence function estimation. Firstly, we compute the generalized fisher information $G(\boldsymbol{\theta})$ on all tunable parameter blocks (LoRA blocks on LoRA-tuned models); Secondly, we compute the inverse of the damped GFIM $(G(\boldsymbol{\theta}) + \lambda I_d)$ with Schulz's iterations (Equation 7); Last, we compute the influence score with cached validation gradient $\boldsymbol{v}$ and the *unflattened* gradient on each training sample, i.e. $\mathcal{I}_{\text{HYPERINF}}(\boldsymbol{x}_k, y_k)$ (Equation 5). We provide the detailed pseudo-code in the Appendix (Algo. 2).

**Complexity Analysis.** Compared to the original influence function formulation in Equation 1, the generalized fisher information matrix $G(\boldsymbol{\theta}^\star) \in \mathbb{R}^{d \times d}$ reduces the memory complexity from $O(r^2 d^2)$ to $O(d^2)$. On computation complexity of Hessian-gradient product, the matrix multiplication between $(G(\boldsymbol{\theta}^\star) + \lambda I_d)^{-1} \in \mathbb{R}^{d \times d}$ and $\boldsymbol{g}_k \in \mathbb{R}^{d \times r}$ only requires $O(rd^2)$ FLOPS, instead of $O(r^2 d^2)$ with flattened gradient vectors. Specifically, with LoRA rank $r = 16$, HYPERINF only requires $0.39\%$ memory complexity and $6.25\%$ computations comparing to original Hessian-vector product operations. We include the complexity comparison to other existing approximation methods in Table 1, where HYPERINF with GFIM showcases outstanding memory and computation efficiencies. In addition, we report the time costs for Hessian inverse-vector product in subsection D.1, where HYPERINF demonstrates superior efficiency on GPU. It underscores the superior compatibility of HYPERINF with modern GPU computations.

---

**Algorithm 1** Matrix Inverse Approximation via Schulz's Iterations

---

**Require:** A matrix $A$ needed to be computed for its inverse, an initial guess $X_0 \approx A^{-1}$, a maximum iteration number $N_{\text{iter}}$.
    **for** $t \in [N_{\text{iter}}]$ **do**
        Iteratively update $X_t = X_{t-1}(2I - AX_{t-1})$
    **end for**
    **return** The final approximation $A^{-1} \leftarrow X_{N_{\text{iter}}}$

---

## 4 SYNTHETIC CONVERGENCE TEST OF MATRIX INVERSE APPROXIMATION

**Setup.** We first examine the accuracy and stability of Schulz's algorithm on matrix inverse approximation by a convergence test. Specifically, to simulate the FIM matrix in the influence function $A = (G(\boldsymbol{\theta}^\star) + \lambda I_d)$ on a training set with scale $|\mathcal{D}^{\text{train}}| = N$ and model with number of parameters as $d$, we construct $M = \frac{1}{N} \sum_{i=1}^{N} s_i s_i^\top + \lambda I \in \mathbb{R}^{d \times d}$ by randomly generating $s_i \in \mathbb{R}^d$. We then compute the exact value of $M^{-1} \in \mathbb{R}^{d \times d}$ and the approximated value $\tilde{M}^{-1}$ using DATAINF and Schulz's algorithm. For LISSA, since it directly approximates the inverted matrix-vector product, we randomly generate another vector $\boldsymbol{v} \in \mathbb{R}^d$ and compute the exact value of the matrix-vector product $Q = M^{-1}\boldsymbol{v} \in \mathbb{R}^d$ as the target. We denote the approximated value from LISSA as $\tilde{Q}$. For all the methods, we measure the error as the Frobenius norm of the matrix $\|Q - \tilde{Q}\|_F$, where $\tilde{Q} = \tilde{M}^{-1}\boldsymbol{v}$ for DATAINF and HYPERINF. We run the convergence test across various $d \in \{512, 1024, 2048, 4096\}$ and $N \in \{200, 800, 6400, 12800\}$, emulating different scales of model and amount of data samples respectively. In all settings, the dampling factor $\lambda$ is set as $0.01$. The initialization for iterative methods is set as $X_0 = 5e^{-4} I_d$. We provide more results with matrices from various distributions in Appendix H, which demonstrates the similar pattern as in Figure 1.

**HYPERINF solves matrix-inversion approximation with great convergence performance.** We present the results from the synthetic experiments in Figure 1, where HYPERINF with Schulz's algorithm demonstrates a remarkable accuracy and stability compared to the other two methods. Specifically, on high-dimensional matrices $M$ with large $d$, both LISSA and DATAINF tend to diverge with increasing approximation errors. For LISSA, the error would not converge but explode exponentially according to the number of iterations. Even when applying on a small dimension of matrix with $N = 200$, LISSA is not able to give an accurate approximation with a large error rate $\sim 10^5$. This might comes from the sensitivity of LISSA algorithm to the initialization conditions, which could be hard to tune when apply on large-scale models. In comparison, HYPERINF with Schulz's algorithm could always converge to a low error rate within finite iterations across all scales of $d$ and $N$. It implies that our proposed HYPERINF could consistently achieve a satisfying accuracy on large-scale models and datasets, while both LISSA and DATAINF could significantly diverge from the exact value.

## 5 INFLUENCE FUNCTION APPROXIMATION ON LARGE-SCALE MODELS

In this section, we further apply HYPERINF on influence function approximation on large-scale foundation models and demonstrate its effectiveness on various data attribution tasks. We compare HYPERINF with two existing baseline methods LISSA (Agarwal et al., 2017) and DATAINF (Kwon et al., 2024), as well as the Hessian-free method TRACIN, which replaces the second-order term $H^{-1}$ in Equation 1 with the identity matrix $I_d$ (Pruthi et al., 2020). Across all mislabeled data

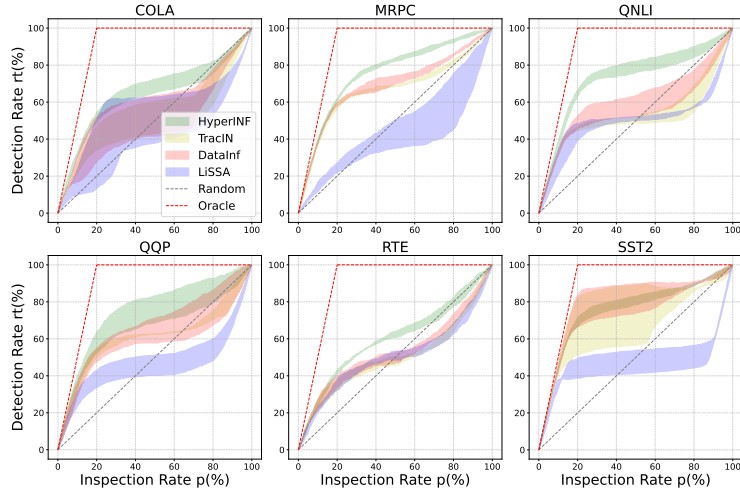

Figure 2: **Mislabeled Data Detection across the GLUE Benchmark with rank $r = 16$ for rsLoRA finetuning.** HYPERINF significantly improve the detection rate ($rt$) according to the inspection rate ($p$) above all baselines, while LISSA performs barely better than the random guess. The dotted lines denote the detection rates from **Random Guess** and **Oracle**, which is the best possible accuracy at each inspection rate. For each method, we run the experiments with 3 random seeds and report the detection rate with 95% confidence intervals.

detection, data selection for LLM fintuning and VLM pretraining, HYPERINF shows promising performance compared to all baseline methods.

## 5.1 MISLABELED DATA DETECTION

We first apply HYPERINF on the mislabeled data detection task following (Koh & Liang, 2020; Yang et al., 2024; Kwon et al., 2024). We construct a corrupted dataset by flipping the label of 20% randomly sampled data points, which is considered as the *mislabeled subset*. After fine-tuning the model on the corrupted training dataset, we rank all data points according to their influence scores from HYPERINF, LISSA and DATAINF respectively and then identify the top-$p$% samples with the highest scores as the mislabeled ones. We define $p$ as the *inspection rate*. Denote the real mislabeled subset as $D_{mis}$ and the identified top-$p$% percentage subset using influence function as $\tilde{D}(p)$, the detection ratio $rt(p)$ can then be measured as the *recall* between $D_{mis}$ and $\tilde{D}(p)$:

$$rt(p) = \frac{|D_{mis} \cap \tilde{D}(p)|}{|D_{mis}|} \in [0, \min(p/20, 1.0)] \tag{8}$$

We assess the mislabeled data detection accuracy according to the detection ratio $rt$ with respect to the inspection rate $p$. We run the experiments across six tasks in the GLUE benchmark (Wang et al., 2019a) with the `Roberta-large` model. We finetune the pretrained `Roberta-large` checkpoint on each corrupted training set using rsLoRA (Kalajdzievski, 2023), a rank-stabilized variant of LoRA (Hu et al., 2021). We provide more implementation details, ablations with various LoRA ranks $r$ and complexity analysis in Appendix D.

**Results.** According to Figure 2, HYPERINF outperforms all baselines on 5 out of 6 tasks with better accuracy and less variance. On SST2, the accuracy of HYPERINF is comparable to DATAINF and TRACIN method while the variance is largely reduced when applying HYPERINF. In contrast, we find that LISSA does not perform well on the mislabeled data detection task: on most of the tasks, the $rt$-$p$ curve approaches linear or horizontal, which indicates LISSA is barely better than the random guess in identifying toxic data points. Additionally, with the low-rank formulation from GFIM, HYPERINF achieves a remarkable efficiency comparable to all the other baselines using GPU computing (subsection D.1).

**Comparison between HYPERINF with GFIM and FIM.** It is worth noting that HYPERINF with GFIM does not lead to performance degradation compared to FIM. According to Figure 5, HYPERINF with GFIM could consistently achieve comparable or better performance than HYPERINF with FIM, while being $(1/r)^3$ more efficient in computation and $(1/r)^2$ in memory (Table 1).

## 5.2 DATA SELECTION FOR LLM FINETUNING

We further manifest the effectiveness of HYPERINF on data selection tasks for LLM finetuning (Pruthi et al., 2020; Kwon et al., 2024; Xia et al., 2024; Albalak et al., 2024). Given a downstream task, we aim to select the high-quality and most relevant data points from the training set which yields a better accuracy on the held-out test set. Specifically, we fine-tune a pretrained `Llama2-7B`[4] checkpoint (Touvron et al., 2023) on four reasoning tasks: QASC (Khot et al., 2020), HellaSwag (Zellers et al., 2019), PIQA (Bisk et al., 2020) and LogiQA (Liu et al., 2020). We consider both sparse (LoRA) and dense finetuning strategies. When applying LoRA, we start with a warmup run on the training set for 1 epoch to prevent using gradients from randomly initialized LoRA modules. We apply LoRA with rank $r = 64$. We compute influence scores from HYPERINF, DATAINF, LISSA and TRACIN and select the top-$k\%$ ($k = 5, 20$) datapoints with the lowest (i.e. *largest negative*) scores respectively. We continually train the model after warmup run using the selected data points. For dense finetuning, we use the gradients from the last transformer block to compute influence scores, which is observed to be the most influential layer within the autoregressive language model architecture (Men et al., 2024). We report the accuracy of the finetuned model evaluated on the held-out test set. We include more implementation details in Appendix E. The model is tuned for $N = 5$ (resp. $N = 3$) epochs on LoRA (resp. dense) finetuning. We also compare to training the model on the full dataset for $N = 1$ epoch.

**Results on LoRA finetuning.** According to Table 2, HYPERINF achieves the best performance comparing to other baselines. Notably, with 5% finetuning datapoints selected by HYPERINF, the reasoning accuracy outperforms the train with the full dataset, which requires 20× data samples and 4× FLOPs. With 20% HYPERINF-selected data points, HYPERINF greatly improves the accuracy by 2.0% above the random selection baseline.

**Results on dense finetuning.** Although the theoretical analysis in Theorem 3.1 is inspired by LoRA finetuning context, we show that data selection by HYPERINF also significantly benefits dense finetuning. According to Table 3, with 5%, 20%, 40% selected data points, HYPERINF consistently improves the reasoning accuracy across all tasks above the random baseline. In contrast, all three baselines could lead to degradation when selecting a small portion of data points ($5, 20\%$). Compared to training on the full dataset (1 epoch), using 40% HYPERINF-selected samples improves the average accuracy by 12.9%, which also performs other baselines by a large margin.

Table 2: Evaluation accuracies (%) for LLM data selection with *LoRA finetuning*. The best results are **Bolded** and the second-best are Underlined. On average, HYPERINF shows the larger improvements as $k$ increases and performs better than all other baselines. The $\uparrow$ ($\downarrow$) indicates the improvement (degradation) compared to the Random baseline.

| Method (*LoRA*) ($k\%$) | | Random | DATAINF | LISSA | TRACIN | HYPERINF |
|---|---|---|---|---|---|---|
| QASC | 5% | **14.0** | 12.7 | 10.6 | 12 | 12.9 |
| | 20% | 16.2 | 18.7 | 16.7 | 16.3 | **19.7** |
| | 100% | 14.1 | - | - | - | - |
| HellaSwag | 5% | 89.4 | 88.9 | 88.5 | 88.5 | **89.6** |
| | 20% | 88.7 | **89.8** | 89.5 | 89.3 | 89.7 |
| | 100% | 91.7 | - | - | - | - |
| PIQA | 5% | 51.3 | 53.7 | 52.9 | 52.9 | **54.1** |
| | 20% | 52.6 | 52.7 | 55.6 | 54.8 | **56.0** |
| | 100% | 50.6 | - | - | - | - |
| LogiQA | 5% | 27.0 | **28.7** | 25.4 | 24.8 | 28.0 |
| | 20% | 26.8 | **27.0** | 25.6 | **27.0** | **27.0** |
| | 100% | 27.6 | - | - | - | - |
| Average | 5% | 45.4 | 46.0$_{(0.6\uparrow)}$ | 44.4$_{(1.0\downarrow)}$ | 44.6$_{(0.8\downarrow)}$ | **46.2**$_{(0.8\uparrow)}$ |
| | 20% | 46.1 | 47.1$_{(1.0\uparrow)}$ | 46.9$_{(0.8\uparrow)}$ | 46.9$_{(0.8\uparrow)}$ | **48.1**$_{(2.0\uparrow)}$ |
| | 100% | 46.0 | - | - | - | - |

[4] https://huggingface.co/meta-llama/Llama-2-7b-hf

## 5.3 DATA SELECTION FOR VLM PRETRAINING

Inspired by the promising performance of HYPERINF on large-scale models and datasets, we further consider to apply it on multimodal instruct-tuning data selection for Vision-Language Model (VLM) pretraining (Liu et al., 2023c; Bai et al., 2023; Chen et al., 2023; Karamcheti et al., 2024).

Following LLaVa (Liu et al., 2023c), we adopt the commonly used VLM architecture which consists of three components: a vision backbone $V_\phi$, a projector $F_\psi$ and a language backbone $LM_\theta$. Both the vision and language backbones are pre-trained, while the projector is randomly initialized. We follow the auto-regressive training paradigm of vision-language models using multimodal instruct-tuning datasets represented as $(x_{\text{img}}, x_{\text{text}}) \in D_{vlm}$. In our experiments, we apply CLIP ViT-Large (Radford et al., 2021) with a patch size of 14 and input resolution of 336px as the vision backbone and Llama2-7B (Touvron et al., 2023) as the language backbone. For the projector $F_\psi$, we initialize a two-layer GELU-MLP (Hendrycks & Gimpel, 2023). Along the suggested setting from Karamcheti et al. (2024), we freeze the vision backbone $V_\phi$ throughout the entire training process while only tuning the projector $F_\psi$ and the language backbone $LM_\theta$. We provide more implementation details in Appendix F.1.

**Setup.** We adopt the two-phase pretraining scheme following LLaVa (Liu et al., 2023c). In the *alignment phase*, we tune the projector $F_\psi$ and LoRA modules of the language backbone on a separate alignment dataset (Karamcheti et al., 2024). For the second instruct-tuning phase, we select the most influential data samples from a large generic multimodal instruct-tuning dataset consisting of 665K datapoints (Karamcheti et al., 2024). We compute the influence score utilizing the gradients from the projector and LoRA modules then select the top-$k\%$ ($k = 5\%, 20\%$) subset with the lowest (i.e. *largest negative*) scores. We train the VLM on the selected instruct-tuning subsets for one epoch and evaluate the model's performance on four cross-modal reasoning tasks: VQAv2 (Goyal et al., 2017), GQA (Hudson & Manning, 2019), POPE (Li et al., 2023) and Text-VQA (Singh et al., 2019). We provide more details on the dataset and implementation in Appendix F.2 and F.3.

**Results.** We present the downstream accuracies across four reasoning tasks in Table 4. On average, HYPERINF consistently outperforms all the other data selection methods and achieves a 2.3% improvement above the random baseline with 20% selected subset. In contrast, with 5% selected data points, LISSA shows a large (8%) performance degradation because of the lack of accurate second-order information.

Table 3: Evaluation accuracies (%) for LLM data selection with *dense finetuning*. The best results are **Bolded** and the second-best are Underlined. On average, HYPERINF could outperform the Random baseline while the other methods fail when the selection ratio $k$ is small. The ↑ (↓) indicates the improvement (degradation) compared to the Random baseline.

| Method (*dense*) ($k\%$) | | Random | DATAINF | LISSA | TRACIN | HYPERINF |
|---|---|---|---|---|---|---|
| QASC | 5% | 11.3 | 12.5 | 11.2 | 11.4 | **14.3** |
| | 20% | 13.3 | **22.2** | 11.7 | 11.0 | 15.0 |
| | 40% | 18.1 | 35.6 | 13.2 | 40.1 | **56.1** |
| | 100% | 11.9 | - | - | - | - |
| HellaSwag | 5% | 71.5 | 70.8 | 70.6 | 72.5 | **81.3** |
| | 20% | **84.7** | 82.8 | 83.8 | 82.6 | 83.2 |
| | 40% | 86.0 | 87.8 | **89.0** | 88.9 | 87.0 |
| | 100% | 92.4 | - | - | - | - |
| PIQA | 5% | 46.5 | 42.3 | 48.7 | 47.8 | **53.2** |
| | 20% | 53.2 | 55.0 | 52.8 | **57.3** | 57.0 |
| | 40% | 55.0 | 60.8 | **60.9** | 57.1 | 58.0 |
| | 100% | 51.0 | - | - | - | - |
| LogiQA | 5% | 25.5 | 25.0 | 27.2 | 25.4 | **28.3** |
| | 20% | 28.6 | 22.3 | 26.4 | 27.4 | **30.2** |
| | 40% | 30.6 | 28.2 | 34.3 | 33.2 | **40.1** |
| | 100% | 27.0 | - | - | - | - |
| Average | 5% | 38.7 | 37.6(1.1↓) | 39.4(0.7↑) | 39.3(0.6↑) | **44.3**(5.6↑) |
| | 20% | 44.9 | 45.6(0.7↑) | 43.7(1.2↓) | 44.6(0.3↓) | **46.4**(1.5↑) |
| | 40% | 47.4 | 53.1(5.7↑) | 49.4(2.0↑) | 54.8(7.4↑) | **60.3**(12.9↑) |
| | 100% | 45.6 | - | - | - | - |

**Skip alignment in training, not data selection.** (Karamcheti et al., 2024) illustrated from extensive empirical experiments that we can skip the alignment phase in VLM pretraining to achieve compa-

rable performance as the two-phase training. To explore whether it applies to data selection, we directly apply HYPERINF, DATAINF, LISSA and TRACIN before alignment. Since the projector gradients are randomly initialized before the alignment phase, we only use the gradients from the last transformer block in language backbone to compute the influence scores. According to F.4, while the HYPERINF could still bring slight improvement ($0.25 - 1\%$) above random baseline, all the other three methods suffer from a significant degradation ($> 5\% \downarrow$) on the accuracy. We hypothesise that the alignment phase is crucial to learning about the connection between the feature spaces of language and vision backbones, which is indispensable information for VLM pretraining data selection. Therefore, we suggest the practitioners apply data selection after the alignment phase.

Table 4: Downstream evaluation accuracies (%) from VLM instruct-tuning data selection experiments (after cross-modal alignment on Projector and LoRA layers). The best results are **Bolded** and the second-best are Underlined. *Projector+LoRA* means the gradient from both the *Projector* and *LoRA* are used to compute approximated scores. Methods with $> 5\%$ accuracy degradation are marked in Red.

| Method (*Projector+LoRA*) ($k\%$) | | Random | DATAINF | LISSA | TRACIN | HYPERINF |
|---|---|---|---|---|---|---|
| VQAv2 | 5% | 60.2 | **60.7** | 53.2 | 59.2 | 60.3 |
| | 20% | 64.5 | 64.7 | 65.1 | 66.4 | **67.3** |
| GQA | 5% | 42.2 | 42.5 | 35.9 | 43.6 | **45.5** |
| | 20% | 45.5 | 45.1 | 46.3 | 49.8 | **50.5** |
| POPE | 5% | 72.2 | 76.9 | 57.9 | 78.9 | **80.6** |
| | 20% | 83.4 | 84.0 | 82.6 | 84.2 | **84.5** |
| TextVQA | 5% | **32.0** | **32.0** | 27.4 | 26.2 | 26.4 |
| | 20% | 35.8 | 35.9 | 34.3 | 31.7 | **36.1** |
| Average | 5% | 51.6 | 53.0$_{(1.4\uparrow)}$ | 43.6$_{(8.0\downarrow)}$ | 51.9$_{(0.3\uparrow)}$ | **53.2**$_{(1.6\uparrow)}$ |
| | 20% | 57.3 | 57.4$_{(0.1\uparrow)}$ | 57.0$_{(0.3\downarrow)}$ | 58.0$_{(0.7\uparrow)}$ | **59.6**$_{(2.3\uparrow)}$ |

## 6 RELATED WORKS

**Gradient-based Data Attribution Methods.** Assessing the importance of each datapoint based on the model's performance is a widely studied problem. Traditional methods based on Sharpley-value and LOO (leave-one-out) mechanism often need to train numerous models to get a reliable score, which limits their application on large models nor datasets (Ghorbani & Zou, 2019; Jia et al., 2020; Kwon & Zou, 2022; Wang & Jia, 2023). In comparison, by tracing the gradient information from the model, one can value the contribution of each datapoint along the optimization process. Various methods are proposed to assess the data influence tracing first-order gradient (Pruthi et al., 2020). However, those methods risk biasing towards dimensions with larger gradient scales and the uncertainty from stochasticity (Pooladzandi et al., 2022). This could be mitigated by influence function-based methods (Koh & Liang, 2020; Kwon et al., 2024; Agarwal et al., 2017), which leverage the second-order curvature information to balance the uncertainty of the first-order gradients.

**Data Selection for Foundation Models.** High-quality datapoints are shown to improve the base LLM's performance dramatically. Increasing datapoint's quality and diversity can effectively induce the instruction-following ability for large language models (Cao et al., 2024; Chen et al., 2024; Du et al., 2023; Li et al., 2024; Liu et al., 2024). Furthermore, researches on both task-based traditional NLP tasks and open-ended instruction tuning datasets have demonstrated its effectiveness (Longpre et al., 2023a; Zhou et al., 2023; Xu et al., 2023; Wei et al., 2021).

## 7 CONCLUSION

In this work, we propose HYPERINF as an efficient approximation of influence function with accurate second-order information, which leverage generalized fisher information and the Schulz's algorithm. From a convergence test on matrix inversion, we demonstrate the superior accuracy and stability of the Schulz's algorithm comparing to other methods. We further illustrate HYPERINF's efficacy in a range of data attribution applications, including mislabel data detection, data selection for LLM finetuning and VLM pretraining. Remarkably, HYPERINF consistently outperforms all the other baselines, which proves the benefit from an accurate estimation of second-order information.

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

## A    DERIVATIONS OF INFLUENCE FUNCTION AND ITS VARIANTS

### A.1    INFLUENCE FUNCTION

We provide the proof for Influence Function based on the work of Koh & Liang (2020). We have $\boldsymbol{\theta}^\star$ denoted as the minimizer for the empirical risk:

$$R(\boldsymbol{\theta}) := \frac{1}{n} \sum_{i=1}^{n} \ell(y_i, f_{\boldsymbol{\theta}}(\boldsymbol{x}_i)) \tag{9}$$

We also assume that the $R$ is twice-differentiable and strongly convex in $\boldsymbol{\theta}$, therefore:

$$H(\boldsymbol{\theta}) := \nabla_{\boldsymbol{\theta}}^2 R(\boldsymbol{\theta}) = \nabla_{\boldsymbol{\theta}}^2 \left( \frac{1}{n} \sum_{i=1}^{n} \ell(y_i, f_{\boldsymbol{\theta}}(\boldsymbol{x}_i)) \right) \tag{10}$$

exists and is positive definite. Then upweighing the contribution of the $k^{th}$ datapoint, we have:

$$\boldsymbol{\theta}^{(k)}(\epsilon) := \arg \min_{\boldsymbol{\theta} \in \Theta} \frac{1}{n} \sum_{i=1}^{n} \ell\left(y_i, f_{\boldsymbol{\theta}}(\boldsymbol{x}_i)\right) + \epsilon \ell\left(y_k, f_{\boldsymbol{\theta}}(\boldsymbol{x}_k)\right) \tag{11}$$

$$= \arg \min_{\boldsymbol{\theta} \in \Theta} R(\boldsymbol{\theta}) + \epsilon \ell(\boldsymbol{x}_k, \boldsymbol{\theta}) \tag{12}$$

Define the change of the parameter $\Delta_\epsilon := \boldsymbol{\theta}^{(k)}(\epsilon) - \boldsymbol{\theta}^\star$ and notice that $\boldsymbol{\theta}^\star$ does not depend on $\epsilon$, the quantity we want to compute in Equation 1 can be re-written as:

$$\frac{d\boldsymbol{\theta}^{(k)}}{d\varepsilon} = \frac{d\Delta_\epsilon}{d\varepsilon} \tag{13}$$

From previous definition, $\boldsymbol{\theta}^{(k)}(\epsilon)$ is the minimizer for Equation 12, therefore we have the first-order optimality condition:

$$\nabla_{\boldsymbol{\theta}} R(\boldsymbol{\theta}^{(k)}(\epsilon)) + \epsilon \nabla_{\boldsymbol{\theta}} \ell(\boldsymbol{x}_k, \boldsymbol{\theta}^{(k)}(\epsilon)) = 0 \tag{14}$$

We then perform the first-order Taylor expansion of the left-hand side since $\boldsymbol{\theta}^{(k)}(\epsilon) \to \boldsymbol{\theta}^\star$ as $\varepsilon \to 0$:

$$0 \approx [\nabla_{\boldsymbol{\theta}} R(\boldsymbol{\theta}^\star) + \epsilon \nabla_{\boldsymbol{\theta}} \ell(\boldsymbol{x}_k, \boldsymbol{\theta}^\star)] + [\nabla_{\boldsymbol{\theta}}^2 R(\boldsymbol{\theta}^\star) + \epsilon \nabla_{\boldsymbol{\theta}}^2 \ell(\boldsymbol{x}_k, \boldsymbol{\theta}^\star)]\Delta_\epsilon \tag{15}$$

We can further obtain:

$$\Delta_\epsilon \approx -[\nabla_{\boldsymbol{\theta}}^2 R(\boldsymbol{\theta}^\star) + \epsilon \nabla_{\boldsymbol{\theta}}^2 \ell(\boldsymbol{x}_k, \boldsymbol{\theta}^\star)]^{-1}[\nabla_{\boldsymbol{\theta}} R(\boldsymbol{\theta}^\star) + \epsilon \nabla_{\boldsymbol{\theta}} \ell(\boldsymbol{x}_k, \boldsymbol{\theta}^\star)] \tag{16}$$

Because $\boldsymbol{\theta}^\star$ is the minimizer for $R(\boldsymbol{\theta})$, we plus $\nabla_{\boldsymbol{\theta}} R(\boldsymbol{\theta}^\star) = 0$ and drop the $\epsilon$-term in the first term of the right-hand side in Equation 16:

$$\Delta_\epsilon \approx -[\nabla_{\boldsymbol{\theta}}^2 R(\boldsymbol{\theta}^\star)]^{-1} \nabla_{\boldsymbol{\theta}} \ell(\boldsymbol{x}_k, \boldsymbol{\theta}^\star) \epsilon \tag{17}$$

Lastly, combining Equation 10 and Equation 13 we can get:

$$\left. \frac{d\boldsymbol{\theta}^{(k)}}{d\varepsilon} \right|_{\varepsilon=0} = -H\left(\boldsymbol{\theta}^\star\right)^{-1} \nabla_{\boldsymbol{\theta}} \ell_k \tag{18}$$

### A.2    INFLUENCE FUNCTION ON VALIDATION LOSS

In particular, the influence of the upweighing datapoint $(\boldsymbol{x}_k, y_k)$ on the loss at a validation datapoint $(\boldsymbol{x}_j^{\text{val}}, y_j^{\text{val}})$ also has a closed-form formula:

$$\mathcal{I}_{\boldsymbol{x}_j^{\text{val}}, y_j^{\text{val}}}(\boldsymbol{x}_k, y_k) := \left. \frac{d\ell(\boldsymbol{x}_j^{\text{val}}, \boldsymbol{\theta}^{(k)}(\epsilon))}{d\varepsilon} \right|_{\varepsilon=0} \tag{19}$$

$$= \nabla_{\boldsymbol{\theta}} \ell(\boldsymbol{x}_j^{\text{val}}, \boldsymbol{\theta}^\star)^\top \left. \frac{d\boldsymbol{\theta}^{(k)}}{d\varepsilon} \right|_{\varepsilon=0} \tag{20}$$

$$= -\nabla_{\boldsymbol{\theta}} \ell(\boldsymbol{x}_j^{\text{val}}, \boldsymbol{\theta}^\star)^\top H\left(\boldsymbol{\theta}^\star\right)^{-1} \nabla_{\boldsymbol{\theta}} \ell_k \tag{21}$$

Therefore, when we want to evaluate the influence on the whole validation dataset, we can get a similar formula:

$$\mathcal{I}(\boldsymbol{x}_k, y_k) = - \left( \frac{1}{m} \sum_{i=1}^{m} \nabla_{\boldsymbol{\theta}} \ell(y_i^{\text{val}}, f_{\boldsymbol{\theta}}(\boldsymbol{x}_i^{\text{val}}))|_{\boldsymbol{\theta}=\boldsymbol{\theta}^*} \right)^\top H\left(\boldsymbol{\theta}^\star\right)^{-1} \nabla_{\boldsymbol{\theta}} \ell_k \tag{22}$$

### A.3 FULL DERIVATION OF DATAINF

Kwon et al. (2024) proposed a closed-form approximation of the Hessian inverse, which greatly improves the computation efficiency. Firstly, following George et al. (2021), when applying the negative log-likelihood loss function $\ell(y, f_{\boldsymbol{\theta}}(x)) = -\log p(y|f_{\boldsymbol{\theta}}(\boldsymbol{x}))$, the second-order Hessian is equivalent to the Fisher Information Matrix (FIM) **in expectation** (Bartlett, 1953), which only involves first-order computations. Consequently, Kwon et al. (2024) approximate the Hessian inverse leveraging the Sherman-Morrison formula [5]:

$$H(\boldsymbol{\theta})^{-1} \approx \left( \frac{1}{n} \sum_{i=1}^{n} \nabla_{\boldsymbol{\theta}}^2 \ell_i + \lambda I_d \right)^{-1} \approx (G(\boldsymbol{\theta}) + \lambda I_d)^{-1} \rightarrow \textit{Approximation with FIM}$$

$$\approx \frac{1}{n} \sum_{i=1}^{n} \left( \nabla_{\boldsymbol{\theta}} \ell_i \nabla_{\boldsymbol{\theta}} \ell_i^{\top} + \lambda I_d \right)^{-1} \rightarrow \textit{Reverse the order of summation and inverse} \quad (23)$$

$$\approx \frac{1}{n\lambda} \sum_{i=1}^{n} \left( I_d - \frac{\nabla_{\boldsymbol{\theta}} \ell_i \nabla_{\boldsymbol{\theta}} \ell_i^{\top}}{\lambda + \nabla_{\boldsymbol{\theta}} \ell_i^{\top} \nabla_{\boldsymbol{\theta}} \ell_i} \right) \rightarrow \textit{Sherman-Morrison formula} \quad (24)$$

where $G(\boldsymbol{\theta}) := \frac{1}{n} \sum_{i=1}^{n} \nabla_{\boldsymbol{\theta}} \ell_i \nabla_{\boldsymbol{\theta}} \ell_i^{\top}$ stands for the Fisher Information Matrix (FIM). While the computation complexity of Equation 24 is reduced to $\mathcal{O}(d)$, in compromise, the reverse-order operation Equation 23 incurs a $\mathcal{O}(d^2)$ error (Kwon et al., 2024). When applying to large-scale models, it could risk a large approximation error.

---

[5]For simplicity, we denote $\ell_i := \ell(y_i, f_{\boldsymbol{\theta}}(\boldsymbol{x}_i))$

## B  PSEUDO CODE FOR HYPERINF

We provide the complete pseudo algorithm using HYPERINF in Algorithm (2) to compute influence function for each datapoint in training set $\mathcal{D}^{\text{train}}$ according to the impact on the validation set $\mathcal{D}^{\text{val}}$.

---

**Algorithm 2** Influence Score computed by HYPERINF

---

**Require:** A training dataset $\mathcal{D}^{(\text{train})} = \{(x_i, y_i)\}_{i=1}^n$, a validation dataset $\mathcal{D}^{(\text{val})} = \{(x_i^{(\text{val})}, y_i^{(\text{val})})\}_{i=1}^m$, an objective function $\ell$, a deep neural network $f_\theta(x) = f_{\theta_L} \circ f_{\theta_{L-1}} \circ ... \circ f_{\theta_1}(x)$, where $\theta = \{\theta_1, ..., \theta_L\}$ and $\theta_l \in \mathbb{R}^{d_l}$ for $l \in [L]$, HYPERINF's initial guess $X_{0,l}$ for $l \in [L]$, HYPERINF's iteration number $N_{\text{iter}}$.

**Ensure:** Influence Score for each training data point: $\mathcal{I}_{\text{HYPERINF}}(x_k, y_k)$ for $k = 1, ..., n$.

# Step 1: Compute the first-order gradients from validation datasets
**for** $l \in [L]$ **do**
    **for** $i \in [m]$ **do**
        Compute $\nabla_{\theta_l} \ell(y_i^{(\text{val})}, f_\theta(x_i^{(\text{val})})) \in \mathbb{R}^{d_l \times r}$, unflattened gradient
    **end for**
    Compute $v_l := \frac{1}{m} \sum_{i=1}^m \nabla_{\theta_l} \ell(y_i^{(\text{val})}, f_\theta(x_i^{(\text{val})}))$
**end for**

# Step 2: Compute the inversion using Schulz's method
**for** $l \in [L]$ **do**
    **for** $i \in [n]$ **do**
        Compute $\nabla_{\theta_l} \ell(y_i, f_\theta(x_i)) \in \mathbb{R}^{d_l \times r}$, unflattened gradient
    **end for**
    Compute $\epsilon_l := 0.1 \times (n d_l)^{-1} \sum_{i=1}^n \nabla_{\theta_l} \ell(y_i, f_\theta(x_i)) \cdot \nabla_{\theta_l} \ell(y_i, f_\theta(x_i))$
    Compute $A_l := G_l(\theta) + \epsilon_l I_{d_l}$
    Compute approximated inversion for $A_l$: $\hat{A_l}^{-1} \leftarrow \text{SCHULZ\_INVERSE}(A_l, X_{0,l}, N_{\text{iter}})$
    Compute the Hessian-Vector Product: $h_l \leftarrow v_l^\top \hat{A_l}^{-1} \in \mathbb{R}^{r \times d_l}$
**end for**

# Step 3: Compute the Influence Score
**for** $k \in [n]$ **do**
    $\mathcal{I}_{\text{HYPERINF}}(x_k, y_k) \leftarrow - \sum_{l=1}^L [h_l \nabla_{\theta_l} \ell(y_k, f_\theta(x_k))]$
**end for**

# Function to compute an inversion of a matrix via Schulz's method
**procedure** SCHULZ\_INVERSE($A, X_0, N_{\text{iter}}$)
    # **Input**: A matrix $A$ needed to be computed for its inverse, an initial guess $X_0$ for $A^{-1}$, a maximum iteration number $N_{\text{iter}}$.
    # **Output**: The final approximation $X_{N_{\text{iter}}}$ for $A^{-1}$.

    **for** $t \in [N_{\text{iter}}]$ **do**
        Iteratively update $X_t = X_{t-1}(2I - AX_{t-1})$
    **end for**
    Get the approximation for $A^{-1} \leftarrow X_{N_{\text{iter}}}$
**end procedure**

---

## C  CONVERGENCE ANALYSIS OF SCHULZ'S METHOD

In this section, we provide convergence analysis of the Schulz's method. We first give the setup with notations:

Let $A \in \mathbb{R}^{n \times n}$ be a non-singular matrix, and $X_k$ be the $k$-th iteration of the Schulz's method, defined as:

$$X_{k+1} = X_k(2I - AX_k), \tag{25}$$

where $X_0$ is the initial approximation of $A^{-1}$. Define the error at $k^{th}$ iteration as: $R_k = I - AX_k$. We provide the proof for the following convergence theorems:

**Theorem C.1.** *The matrix of error $R_k$ satisfies a quadratic relation. I.e.,*

$$R_{k+1} = R_k^2.$$

*Proof.* According to Equation 25, at $k^{th}$ iteration, we have:

$$AX_{k+1} = AX_k(2I - AX_k) = AX_k(I + R_k).$$

Plug into $R_{k+1} = I - AX_{k+1}$, we have,

$$R_{k+1} = I - AX_{k+1} = I - AX_k(I + R_k) = I - AX_k - AX_kR_k.$$

By definition, $R_k = I - AX_k \Rightarrow AX_k = I - R_k$, which gives:

$$R_{k+1} = I - (I - R_k) - (I - R_k)R_k = R_k^2.$$

$\square$

**Theorem C.2.** *The spectral norm of the error decreases quadratically:*

$$\|R_{k+1}\| \leq \|R_k\|^2. \tag{26}$$

*Proof.* Taking norms on both sides:

$$\|R_{k+1}\| = \|R_k^2\|.$$

Applying the submultiplicative property of matrix norms:

$$\|R_k^2\| \leq \|R_k\| \cdot \|R_k\|.$$

Thus we obtain:

$$\|R_{k+1}\| \leq \|R_k\|^2.$$

This proves that the error decreases quadratically with each iteration, provided $\|R_0\| < 1$.  $\square$

**Theorem C.3.** *Given the initial condition that the spectral norm of $R_0 = I - AX_0$ satisfies $\|R_0\| < 1$, then $\lim_{k \to \infty} \|R_k\| \to 0$, $\lim_{k \to \infty} X_k \to A^{-1}$.*

*Proof.* Given $\|R_0\| < 1$, then $\|R_k\|$ satisfies:

$$\|R_k\| \leq \|R_0\|^{2^k}$$

following the above proved iterative relation $\|R_{k+1}\| \leq \|R_k\|^2$. As $k \to \infty$, $\|R_k\| \to 0$ exponentially fast. Consequently, as $k \to \infty$,

$$X_k \to A^{-1}.$$

$\square$

# D  DETAILS FOR MISLABELED DATA DETECTION TASK

**Implementation Details.**  In this task, we choose rank-stabilized LoRA (Kalajdzievski, 2023) instead of original LoRA (Hu et al., 2021), for it corrects the one limitation of LoRA (i.e. the performance did not improve further with increasing rank) by a simply dividing LoRA adapters by the square root of their rank, which unlocks the effectiveness of higher adapter ranks in LoRA.

We conduct mislabeled data detection experiment on six binary classification tasks based on GLUE benchmark (Wang et al., 2019a), which are GLUE-COLA ((Warstadt et al., 2019), detecting whether a sentence is grammatical acceptable) GLUE-MRPC ((Dolan & Brockett, 2005), detecting whether the sentences in the pair are semantically equivalent), GLUE-QNLI ((Rajpurkar et al., 2016), determining whether the context sentence contains the answer to the question), GLUE-QQP[6] (determining whether a pair of questions are semantically equivalent), GLUE-RTE ((Dagan et al., 2006; Bar Haim et al., 2006; Giampiccolo et al., 2007; Bentivogli et al., 2009), detecting the entailment), and GLUE-SST2 ((Socher et al., 2013), predicting the sentiment of a given sentence).

When finetuning the LLM with rsLoRA technique with rank $r = 16$ in Figure 2 and $r = 64$ in Figure 3, we apply the gradients from trainable parameters (i.e. every value and query matrix of the attention layers) to approximate influence functions. We run HYPERINF for 25 iterations and run LiSSA for 10 iterations following the implementation of Kwon et al. (2024). The total number of tunable parameters is $1.6M, 7.3M$ respectively for $r = 16, 64$.

Moreover, We also experiment using the last layer's gradients of `Roberta-large` to detect the mislabeled datapoints. We only tune the last layer of the model on the corrupted training dataset, then compute the influence function based on the last layer's gradients. The results are shown in Figure 4, which indicates that the last layer's gradients can also be a candidate for computing the influence function.

Table 5: Mislabeled Data Detection Rate (%) with $r = 16$.

| Method (*LoRA*) ($k\%$) | | DATAINF | LiSSA | TRACIN | HYPERINF |
|---|---|---|---|---|---|
| COLA | 20% | 39.66 | 32.18 | 40.25 | **51.55** |
|      | 40% | 50.59 | 48.81 | 49.74 | **66.04** |
| MRPC | 20% | 58.52 | 24.46 | 57.75 | **60.89** |
|      | 40% | 68.89 | 37.88 | 67.34 | **79.17** |
| QNLI | 20% | 48.92 | 43.70 | 45.37 | **64.77** |
|      | 40% | 56.51 | 50.18 | 49.51 | **76.66** |
| QQP  | 20% | 51.11 | 38.14 | 52.18 | **57.85** |
|      | 40% | 62.07 | 44.74 | 61.59 | **73.07** |
| RTE  | 20% | 36.74 | 35.07 | 35.14 | **47.90** |
|      | 40% | 47.85 | 47.85 | 45.51 | **57.96** |
| SST2 | 20% | **74.96** | 44.93 | 66.51 | 69.00 |
|      | 40% | **80.5**1 | 46.62 | 71.96 | 78.44 |

---

[6]https://quoradata.quora.com/First-Quora-Dataset-Release-Question-Pairs

**Comparisons between HYPERINF with GFIM and HYPERINF with FIM**    To explore if using GFIM can lead to performance degradation, we compare HYPERINF with GFIM and HYPERINF with FIM. In this experiment, we set rank $r = 8$ since larger ranks (e.g. $r = 16, 32, ...$) would cause the Out-Of-Memory error in FIM. The results are shown in Figure 5, where we do not observe the significantly worse performance in HYPERINF with GFIM, and it performs even better on some datasets than FIM, such as QQP and SST2.

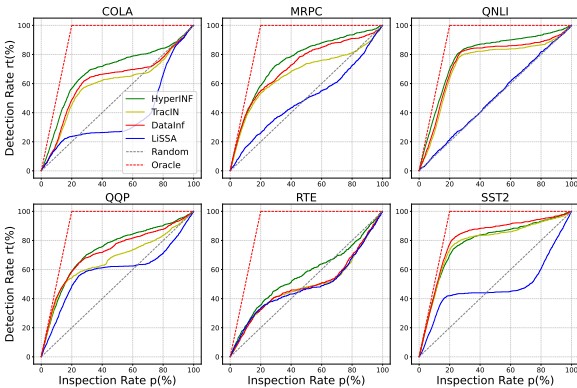

Figure 3: Mislabeled data detection results on GLUE benchmark datasets with rank $r = 64$, $\#params = 7.3M$.

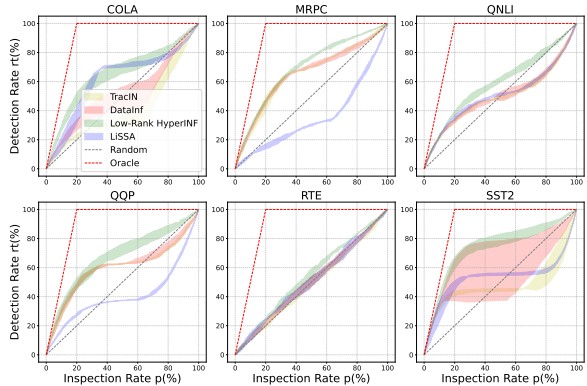

Figure 4: Mislabeled data detection results on GLUE benchmark datasets, where influence function is computed based on the last layer's gradients.

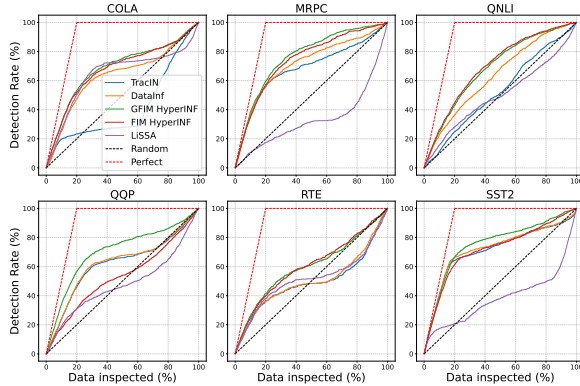

Figure 5: Mislabeled data detection results on GLUE benchmark datasets with rank $r = 8$.

### D.1 ANALYSIS OF COMPLEXITY AND TIME COSTS.

To understand the computation overheads incurred from different data attribution algorithms, we report both time costs on CPU and one Nvidia A100 GPU according to 6 and 7 on two datasets (COLA and MRPC) from the GLUE benchmark. Specifically, we only record the running time for computing the inverse Hessian vector product $v^{\top}G(\theta)$ with different LoRA ranks $r = 1, 2, 4, 8, 16$. We observe that the efficiency of three algorithms ranks largely differently between GPU and CPU. On CPU, DATAINF introduces least time overheads while HYPERINF incurs the most amount of extra time costs. In addition, the time costs from DATAINF and LISSA increase quadratically with LoRA rank $r$ while HYPERINF increase linearly (note that the y-axis is on `log` scale). Alternatively, on one Nvidia A100 GPU, the time costs from all algorithms are almost constant across LoRA ranks, and HYPERINF costs least of time, followed by DATAINF. In comparison, LISSA requires ($\sim 4\times$) more time costs than HYPERINF and DATAINF.

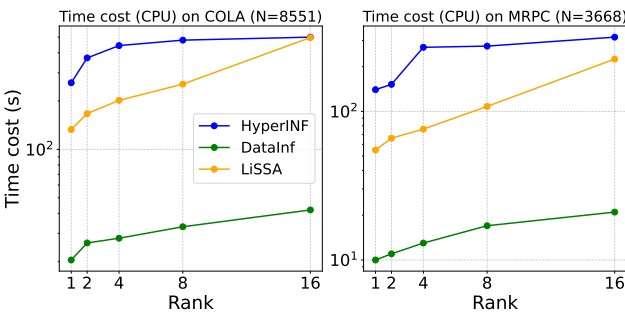

Figure 6: Runtime on CPU for approximating Hessian-vector product using different methods on GLUE-COLA and GLUE-MRPC datasets.

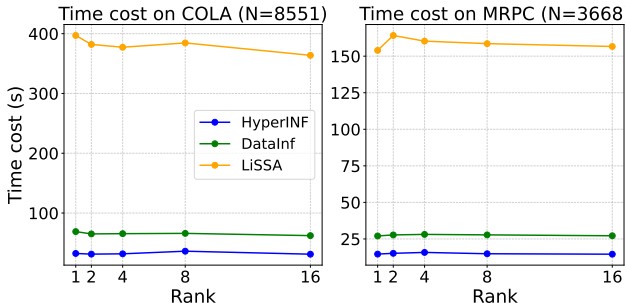

Figure 7: Runtime on GPU for approximating Hessian-vector product using different methods on GLUE-COLA and GLUE-MRPC datasets. HYPERINF takes lowest time costs compared to other methods.

## D.2 CORRELATION WITH LEAVE-ONE-GROUP-OUT (LOGO) SCORES.

The performance of a training data attribution (TDA) algorithm can be assessed by its ability to recover the true Leave-One-Out (LOO) score (Tukey, 1958) The LOO score of a given datapoint $x_i$ is defined as the gap of validation losses of a model before and after removing the certain datapoint. To prevent the large computations incurred from retraining LLMs, we evaluate the TDA algorithms with Leave-One-Group-Out (LOGO). Firstly, we rank all training datapoints according to assigned scores and split them equally into $K$ groups from high to low scores ($K = 5$ in our experiments). On each group of data $C_i$, we iteratively remove $C_i$ and retraining the LLM on the remaining set of data $D_{train}/C_i$. We define the LLM trained on the full training set as $\theta_0$ and the LLM retrained with removing $C_i$ as $\theta_{/C_i}$ Then we measure the LOGO score as:

$$LOGO(C_i) = L(\theta_{/C_i}, D_{val}) - L(\theta_0, D_{val}) \tag{27}$$

If $C_i$ contains high quality datapoints, excluding $C_i$ would hurt the model's performance and lead to an increment of validation loss. Therefore, the LOGO score is proportional to the data quality within the group. In that case, we measure the rank correlation between the average influence score assigned to all groups and the corresponding LOGO scores. We report the spearman rank correlation scores on all four algorithms across six datasets in GLUE benchmark in Table 6. The results demonstrate HYPERINF outperforms all the other baselines on the accuracy of data attribution.

| Method (*LoRA*) | DATAINF | LISSA | TRACIN | HYPERINF |
|---|---|---|---|---|
| COLA | 0.50 | 0.49 | -0.99 | **0.70** |
| MRPC | 0.0 | 0.0 | 0.0 | **0.20** |
| QNLI | -0.40 | -0.30 | -0.60 | **0.10** |
| QQP | 0.30 | 0.49 | -0.30 | **0.70** |
| RTE | 0.60 | 0.60 | 0.40 | **1.00** |
| SST2 | -0.90 | -0.30 | -0.10 | **0.70** |

Table 6: Spearman Rank Correlation.

# E  DATA SELECTION FOR LLM FINETUNING

**Dataset Details.**  We run the experiments on four LLM reasoning tasks: QASC (a question-answering dataset with a focus on sentence composition. It consists of $9,980$ 8-way multiple-choice questions about grade school science) (Khot et al., 2020), HellaSwag (a challenging dataset for evaluating commonsense NLI) (Zellers et al., 2019), PIQA (a dataset introducing the task of physical commonsense reasoning) (Bisk et al., 2020) and LogiQA (is constructed from the logical comprehension problems from publically available questions of the National Civil Servants Examination of China) (Liu et al., 2020). For LogiQA, we use the official validation set as $\mathcal{D}^{val}$ in data selection and use labelled official test set for evaluation; for other three datasets, since the labels for the official test set are not available, we randomly split $20\%$ from the official validation set as $\mathcal{D}^{val}$, and use the rest $80\%$ validation set as the held-out test set.

**Implementation Details.**  For LoRA-finetuning, we follow the same setting as we implement in Mislabeled Data Detection task while setting the rank $r = 64$. The hyperparameters are set as the same as in VLM experiments (Table 7), while the Epoch number is set to 3 for fully-finetuning and 5 for LoRA-finetuning across $k = 5\%, 20\%, 40\%$. When selecting all datapoints (i.e. $k = 100\%$), we finetune it for only 1 epoch.

**Evaluation Statistics.**  We present the detailed statistics of evaluation results in Table 2 and Figure 8 for LoRA-finetuning experiments, and Table 3 and Figure 9 for fully-finetuning experiments. HYPERINF significantly outperforms all baselines.

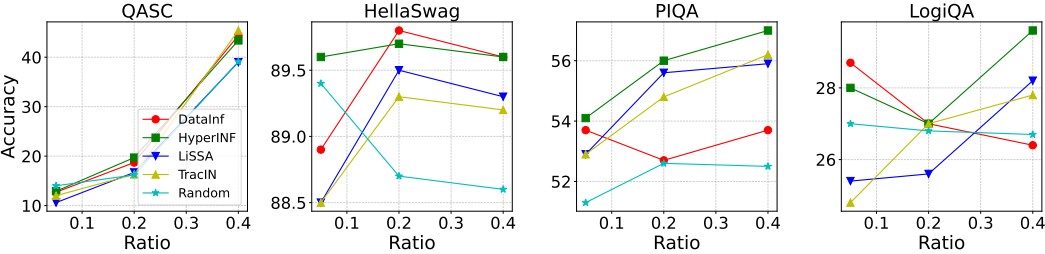

Figure 8: **Evaluation accuracy according to data selection ratio ($k$) for LLM LoRA-finetuning.** HYPERINF greatly improves the reasoning accuracy above other baselines.

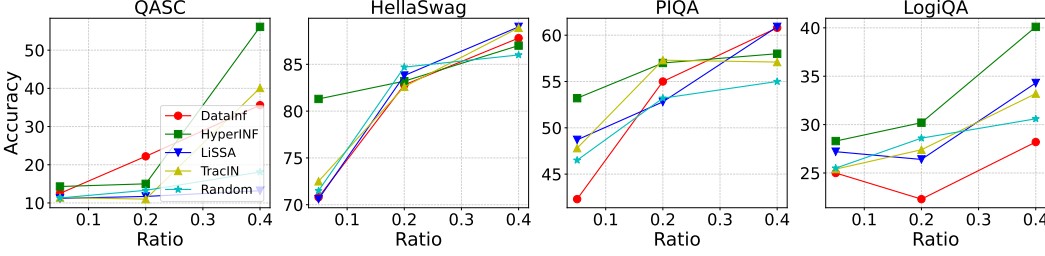

Figure 9: **Evaluation accuracy according to data selection ratio ($k$) for LLM fully-finetuning.** Influence scores are computed based on the gradients of the last layer of LLM. HYPERINF shows significantly better performances above other baselines especially when $k = 5\%$.

## F   Data Selection for VLM Pretraining

### F.1   Details of VLM Architecture and Training Strategy

Following LLaVa (Liu et al., 2023c), we adopt the commonly used VLM architecture which consists of three components: a vision backbone $V_\phi$, a projector $F_\psi$ and a language backbone $LM_\theta$. Both the vision and language backbones are pre-trained, while the projector is randomly initialized and would be tuned through the alignment and instruct-tuning phases using multimodal data (Karamcheti et al., 2024; Liu et al., 2023c; Bai et al., 2023; Chen et al., 2023). We follow the auto-regressive training paradigm of vision-language models, where the images are tokenized into patches (i.e. visual tokens) to fit into the conventional training patterns of language models. Specifically, each datapoint in a multimodal instruct-tuning dataset can be represented as a tuple $(\boldsymbol{x}_{\text{img}}, \boldsymbol{x}_{\text{text}})$. We get a sequence of embeddings of the image patches through the vision backbone $\boldsymbol{p}_{\text{img}} = V_\phi(\boldsymbol{x}_{\text{img}})$ then feed it into the projector to obtain the transformed features $\boldsymbol{e}_{\text{img}} = F_\psi(\boldsymbol{p}_{\text{img}})$. Meanwhile, we have the embeddings from textual tokens as $\boldsymbol{e}_{\text{text}} = LM_\theta(\boldsymbol{x}_{\text{text}})$. We then concatenate the features from both modalities together to conduct next-token predictions. In our experiments, we apply `CLIP ViT-Large` (Radford et al., 2021) with a patch size of $14$ and input resolution of 336px as the vision backbone and `Llama2-7B` (Touvron et al., 2023) as the language backbone. For the projector $F_\psi$, we initialize a two-layer GELU-MLP (Hendrycks & Gimpel, 2023). Along the suggested setting from Karamcheti et al. (2024), we freeze the vision backbone $V_\phi$ throughout the entire training process while only tuning the projector $F_\psi$ and the language backbone $LM_\theta$.

Specifically, we utilize the Prismatic-VLM framework[7] (Karamcheti et al., 2024) to train the VLM. We use 6xA100 80G GPUs to train the model, and the hyperparameters are set as Table 7.

Table 7: Hyperparameters setting for training VLM

| Hyperparameters | Values |
| --- | --- |
| Epoch | 1 |
| Optimizer | AdamW |
| Learning Rate | 2e-5 |
| Weight Decay | 0.1 |
| Max Grad Norm | 1.0 |
| Warmup Ratio | 0.03 |
| Batch Size per GPU | 16 |
| Scheduler | Warmup & Cosine Decay |

### F.2   Details of VLM Dataset

**Instruct-tuning Dataset.**   We follow the work of Karamcheti et al. (2024) and this dataset contains 665K multimodal instruct tuning examples[8]. Liu et al. (2023b) has identified a set of "trigger prompts" for each dataset in the mixture, to induce more capabilities of VLM. The datasets are sourced as follows, where we removed *ShareGPT* (language-only) in our experiments. We split it into a training dataset and a validation dataset as $8:2$ ratio.

*LlaVa Synthetic Data* (158K): A synthetically generated dataset of conversations, fine-grained descriptions, and question-answering data from Liu et al. (2023c), built by prompting GPT-4 (OpenAI et al., 2024) with image captions and object bounding boxes from COCO (Lin et al., 2014).

*Standard VQA Data* (224K): A combination of visual question answering data sourced from the training sets of VQAv2 (general question answering) (Goyal et al., 2017), GQA (spatial and compositional reasoning) (Hudson & Manning, 2019), OK-VQA (reasoning requiring external knowledge) (Marino et al., 2019), and OCR-VQA (reasoning over text/logos in images) (Mishra et al., 2019). LLaVA v1.5 defines the following trigger prompt: "⟨Question⟩? Answer the question using a single word or phrase."

---

[7] `https://github.com/TRI-ML/prismatic-vlms?tab=readme-ov-file`

[8] It can be downloaded following the instructions of `https://github.com/TRI-ML/prismatic-vlms`

*Multiple Choice VQA Data* (50K). Multiple choice visual question answering data sourced from A-OKVQA (requires diverse external knowledge) (Schwenk et al., 2022). LLaVa v1.5 defines the following trigger prompt: "⟨Question⟩? A. ⟨Option A⟩ B. ⟨Option B⟩... Answer with the option's letter from the given choices directly."

*Captioning Data* (22K). Images and captions sourced from TextCaps (images with text/logos) (Sidorov et al., 2020). LLaVa v1.5 defines the following trigger prompt: "Provide a one-sentence caption for the provided image."

*Referring Expression Data* (116K). Referring expression grounding (bounding box prediction) and region captioning data sourced from RefCOCO (Kazemzadeh et al., 2014; Yu et al., 2016) and Visual Genome (Krishna et al., 2016). For bounding box prediction (localization), the model needs to generate normalized bounding box coordinates (as a natural language string). For the localization task, LLaVa v1.5 defines the following trigger prompt: "⟨Referring Expression⟩ Provide the bounding box coordinates of the regionthis sentence describes."

For the inverse task (region caption), LLaVa v1.5 defines a separate trigger prompt: "Provide the bounding box coordinate of the region this sentence describes."

### F.3 DATA SELECTION AFTER CROSS-MODAL ALIGNMENT WITH PROJECTOR AND LORA OF LANGUAGE BACKBONE

**Details of Cross-Modal Alignment.** We keep the same hyperparameter setting as in Table 7 and adopt LoRA to the language backbone. We keep the same LoRA setting in the LLM LoRA-finetuning. In the alignment phase, we tune the projector and LoRA layers while keeping other parts frozen. We use the Vision-Language Alignment dataset (Karamcheti et al., 2024), which consists of 558K (image, caption) pairs, where the caption is a sentence description of the corresponding image. The images are sourced from LAION (Schuhmann et al., 2021), Conceptual Captions (Sharma et al., 2018) and SBU Captions (Ordonez et al., 2011). Considering the limited computation resources, we randomly select 5% datapoints from the alignment dataset for the alignment phase. We leave the larger-scale experiments to future work.

**Details of the Instruct-tuning.** Because of the limited computation resources, we constrain our experiments on 10% of instruct-tuning training dataset used in F.2. We compute the influence function based on the gradients from both Project and LoRA layers, then select $k = 5\%, 20\%, 40\%$ datapoints using various influence function-based methods from the 10% training subset, which is equivalent to 0.5%, 2%, 4% of the original 665K instruct-tuning dataset. In this experiment, we also finetune the projector and LoRA layers of the language backbone and keep other parts frozen.

### F.4 VLM PRETRAINING BEFORE CROSS-MODAL ALIGNMENT

**Setup.** Karamcheti et al. (2024) illustrated from extensive empirical experiments that only applying instruct-tuning can achieve comparable performant pretrained VLMs as the conventional two-phase training (*cross-modal alignment then instruct-tuning*) for LLaVA (Liu et al., 2023c). Thus, we hereby skip the alignment phase in LLaVA (Liu et al., 2023c) and aim to select the most beneficial multi-modal instruct-tuning datapoints for more efficient VLM pretraining (instruct-tuning only). Since the projector is randomly initialized which is not suitable for computing influence function, we use the gradient of the last layer of the pretrained language backbone for HYPERINF and all baselines, to select the datapoints. In this experiment, we compute all instruct-tuning training datapoint's influence score of each method, then select the top-$k\%$ ($k = 20\%, 40\%, 80\%$) subset with the lowest scores. During instruct tuning of this experiment, we tune the projector and the whole language backbone while keeping the vision backbone frozen.

**Results.** We present the evaluation accuracies on four multimodal downstream tasks in Table 8. Notably, when selecting $k = 20\%$ of datapoints, HYPERINF improves the accuracy in average by 7.20% above DATAINF, 8.37% above LISSA and 9.11% above TRACIN. However, we also note that when the selection ratio gets larger ($k > 40\%$), the performance of other baselines will

approach HYPERINF, since the impact from approximation errors on the data ranking is mitigated. Meanwhile, we observe that the random selection is a very strong baseline for all tasks, where only HYPERINF has a small improvement above the random baseline (0.25%) in average accuracy while all the other methods cause a large performance degradation ($> 5\%$). We hypothesize that using pretrained LLM backbone without leveraging cross-modal alignment information may lead to sub-optimal results.

**Evaluation Statistics.** We present detailed statistics for downstream evaluations in Table 8 and Figure 10. HYPERINF greatly improves the accuracies across all tasks above the other data selection baselines, while the random selection is a strong baseline. When selecting 20% subset, HYPERINF is the only method that could outperform random selection according to average accuracy.

Table 8: Downstream evaluation accuracies (%) from VLM instruct-tuning data selection experiments (before cross-modal alignment). The best results are **Bolded** and the second-best are Underlined. The gradient from the last layer of the language backbone is used to compute approximated scores. HYPERINF could outperform the Random baseline while the other methods fail when selection ratios are small. The $\uparrow$ ($\downarrow$) indicates the improvement (degradation) compared to the Random baseline. Methods with $> 5\%$ accuracy degradation are marked in Red.

| Method ($k\%$) | | Random | DATAINF | LISSA | TRACIN | HYPERINF |
|---|---|---|---|---|---|---|
| VQAv2 | 20% | **71.30** | 66.91 | 66.20 | 65.33 | 70.40 |
| | 40% | 74.84 | 75.35 | **75.92** | 75.84 | 75.27 |
| | 60% | 76.29 | 75.35 | **76.99** | 76.95 | 76.89 |
| GQA | 20% | 55.92 | 53.29 | 52.23 | 51.03 | **57.97** |
| | 40% | 59.83 | 60.95 | **62.41** | 61.76 | 61.63 |
| | 60% | 61.49 | 62.97 | 63.11 | 62.62 | **63.35** |
| POPE | 20% | **86.11** | 86.04 | 85.52 | 85.04 | 85.66 |
| | 40% | 86.58 | 85.98 | 86.39 | 86.52 | **86.91** |
| | 60% | **87.00** | 86.63 | 86.40 | 86.99 | 86.92 |
| TextVQA | 20% | 36.20 | 15.50 | 13.10 | 12.70 | **36.50** |
| | 40% | 45.00 | 45.60 | 44.90 | 44.90 | **45.70** |
| | 60% | 47.60 | **49.40** | 48.90 | 49.20 | 49.20 |
| Average | 20% | 62.38 | 55.43$_{(6.95\downarrow)}$ | 54.26$_{(8.12\downarrow)}$ | 53.52$_{(8.86\downarrow)}$ | **62.63**$_{(0.25\uparrow)}$ |
| | 40% | 66.56 | 66.97$_{(0.41\uparrow)}$ | 67.25$_{(0.69\uparrow)}$ | **67.40**$_{(0.84\uparrow)}$ | 67.38$_{(0.82\uparrow)}$ |
| | 60% | 68.09 | 68.59$_{(0.50\uparrow)}$ | 68.85$_{(0.76\uparrow)}$ | 68.94$_{(0.85\uparrow)}$ | **69.09**$_{(1.00\uparrow)}$ |

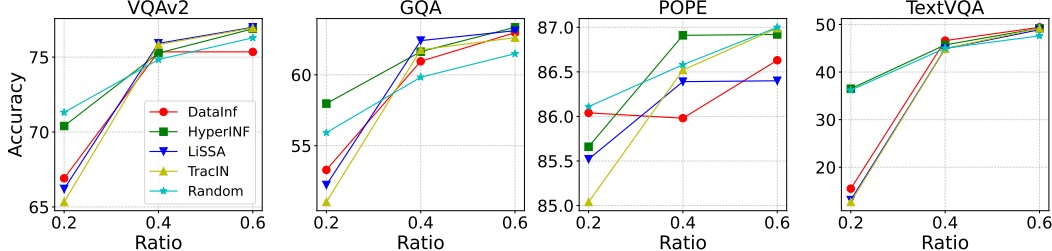

Figure 10: **Downstream evaluation for VLM instruct-tuning data selection (before cross-modal alignment).** HYPERINF benefits the most when selecting a small subset $k = 20\%$, from its accurate approximation of influence function. With $k$ increasing, the performance of other baselines approach HYPERINF, since the impact from approximation errors is mitigated. Random selection is a strong baseline for all data selection methods.

## G    COMPARISON BETWEEN MATRIX INVERSION ALGORITHMS

**Implementation Details.**    In this section, we compare the efficiency of computing inverse of matrices between Schulz's method and other commonly used methods[9], including Gaussian Elimination, Conjugate Gradient, Generalized Minimal Residual method (GMRES) and Faster Gaussian Elimination (i.e. `torch.inverse`). For the iterative methods, we all set the number of iterations to 20 for fair comparisons. We follow the same step in Section. 4 to construct the invertible matrix $M$, and set the dimension of the matrix in different scales: $d \in \{16, 64, 256, 1024, 4096\}$ and $N = 12800$. We use the Frobenius Norm to measure the error between the approximated and true inverse, where we set the Gaussian Elimination as the ground truth. In addition to the error comparison, we also compare the time cost of each method in terms of efficiency aspect. We run the experiments with 3 random seeds and report the average and standard deviation of time costs. All the experiments are done with a single A100 GPU.

**Results.**    The comparisons of error and time cost are shown in Table 9 and Table 10 as well as Figure 11. Schulz achieves a similar error margin as FGE, which is better than CG and GMRES in most cases. Furthermore, Schulz also has the lowest time cost generally in different dimension settings even when $d = 4096$, while other methods observe a significant increase in running time as ranks become larger(especially for Gaussian Elimination, Conjugate Gradient and GMRES). This illustrates the efficiency and stability of HYPERINF since Schulz's method is the main part of our method.

Table 9: Error comparisons among different methods for computing the inverse of the matrix. CG, and FGE denote the Conjugate Gradient and Faster Gaussian Elimination respectively. We reimplemented all the algorithms in `torch` if the original implementation does not support GPU acceleration.

| Matrix Dim | CG | FGE | GMRES | Schulz |
|---|---|---|---|---|
| 16 | 3.5e-10 $\pm$1.2e-10 | 3.0e-11 $\pm$3.1e-12 | 1.3e-10 $\pm$4.2e-11 | 4.2e-11 $\pm$5.1e-12 |
| 64 | 9.7e-10 $\pm$5.2e-11 | 8.7e-11 $\pm$8.6e-12 | 1.6e-10 $\pm$1.7e-11 | 1.4e-10 $\pm$3.9e-12 |
| 256 | 9.9e-9 $\pm$3.6e-10 | 3.9e-10 $\pm$1.1e-11 | 8.9e-10 $\pm$1.3e-10 | 5.4e-10 $\pm$1.3e-11 |
| 1024 | 1.2e-8 $\pm$5.3e-10 | 2.1e-9 $\pm$1.8e-11 | 3.7e-9 $\pm$3.8e-11 | 2.5e-9 $\pm$3.1e-11 |
| 4096 | 1.2e-7 $\pm$5.1e-10 | 2.1e-8 $\pm$1.9e-10 | 1.5e-7 $\pm$7.5e-10 | 2.7e-8 $\pm$2.0e-10 |

Table 10: Time cost (s) comparisons among different methods for computing the inverse of the matrix. GE, CG and FGE denote the Gaussian Elimination, Conjugate Gradient and Faster Gaussian Elimination respectively. We reimplemented all the algorithms in `torch` if the original implementation does not support GPU acceleration.

| Matrix Dim | GE | CG | FGE | GMRES | Schulz |
|---|---|---|---|---|---|
| 16 | 0.04 $\pm$0.02 | 0.11 $\pm$0.005 | 0.02$\pm$0.03 | 0.41$\pm$0.02 | **0.002**$\pm$**0.002** |
| 64 | 0.31 $\pm$0.02 | 0.43$\pm$0.03 | 0.01$\pm$0.01 | 2.27$\pm$0.17 | **0.0008**$\pm$**0.0001** |
| 256 | 2.55$\pm$0.02 | 2.37$\pm$0.11 | **0.001**$\pm$**0.0005** | 12.7$\pm$0.31 | 0.002$\pm$0.002 |
| 1024 | 23.7$\pm$0.10 | 14.6$\pm$0.06 | 0.007$\pm$0.0003 | 77.1 $\pm$0.44 | **0.002** $\pm$**0.002** |
| 4096 | 313.8$\pm$2.29 | 107.9$\pm$5.13 | 0.07$\pm$0.009 | 581.6$\pm$8.15 | **0.001**$\pm$**0.0005** |

---

[9]https://github.com/devzhk/Pytorch-linalg

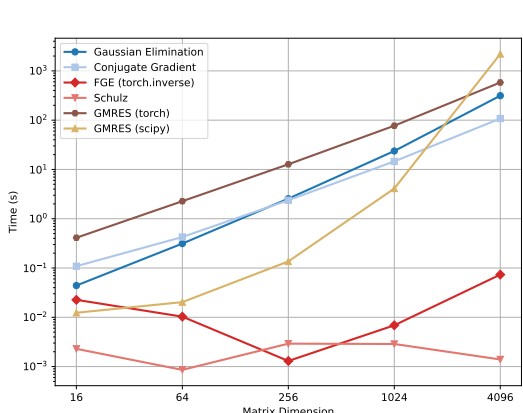

Figure 11: Time cost comparisons among different methods for computing the inverse of the matrix. Schulz presents superior efficiency than other methods.

# H    SUPPLEMENT RESULTS OF CONVERGENCE TEST ON MATRIX INVERSION

We follow the same setting as in section 4 and construct matrices $M = \frac{1}{N}\sum_{i=1}^{N} s_i s_i^\top + \lambda I \in \mathbb{R}^{d \times d}$. To study the convergence with various data distribution and initialization condition, we report the results with $s_i$ and $v$ vectors drawn from 5 difference distributions:

- Each element of $s_i$ and $v$ are drawn from Standard Normal Distribution: $\mathcal{N}(0, 1)$
- Each element of $s_i$ and $v$ are drawn from Normal Distribution: $\mathcal{N}(0.5, 1)$
- Each element of $s_i$ and $v$ are drawn from Normal Distribution: $\mathcal{N}(0, 5)$
- Each element of $s_i$ and $v$ are drawn from Normal Distribution: $\mathcal{N}(0.5, 5)$
- Each element of $s_i$ and $v$ are drawn from Uniform Distribution: $U(0, 1)$

We also include the Neumann Series (which is the same method of LiSSA) and Successive Over Relaxation (SOR) methods to compare. For SOR, the iteration is shown as:

$$X^{(k+1)} = (D - \omega L)^{-1}(\omega U + (1 - \omega)D)X^{(k)} + \omega(D - \omega L)^{-1} \qquad (28)$$

where $D, L, U$ denote the diagonal, lower and upper triangular parts of $M$. $\omega$ is a hyperparameter, when $\omega > 1$ it is overrelaxation, and when $\omega < 1$ it is underrelaxation. We choose $\omega = 0.5, 1.5$ for experiments. To measure the error for all methods, we use the Frobenius norm of the matrix $\|\hat{Q} - Q\|_F$.

**Results.**    The results are shown as Figure 12, Figure 13, Figure 14, Figure 15, and Figure 16. HY-PERINF with Schulz's algorithm demonstrates remarkable stability and convergence performance, which is robust with various data distribution and initial conditions. LISSA only converges in a few circumstances, indicating it's sensitive to the initial condition and matrix distributions. For SOR, only when the data distribution is from $\mathcal{N}(0, 1)$ (see Figure 12 and Figure 13) it can converge in limited circumstances.

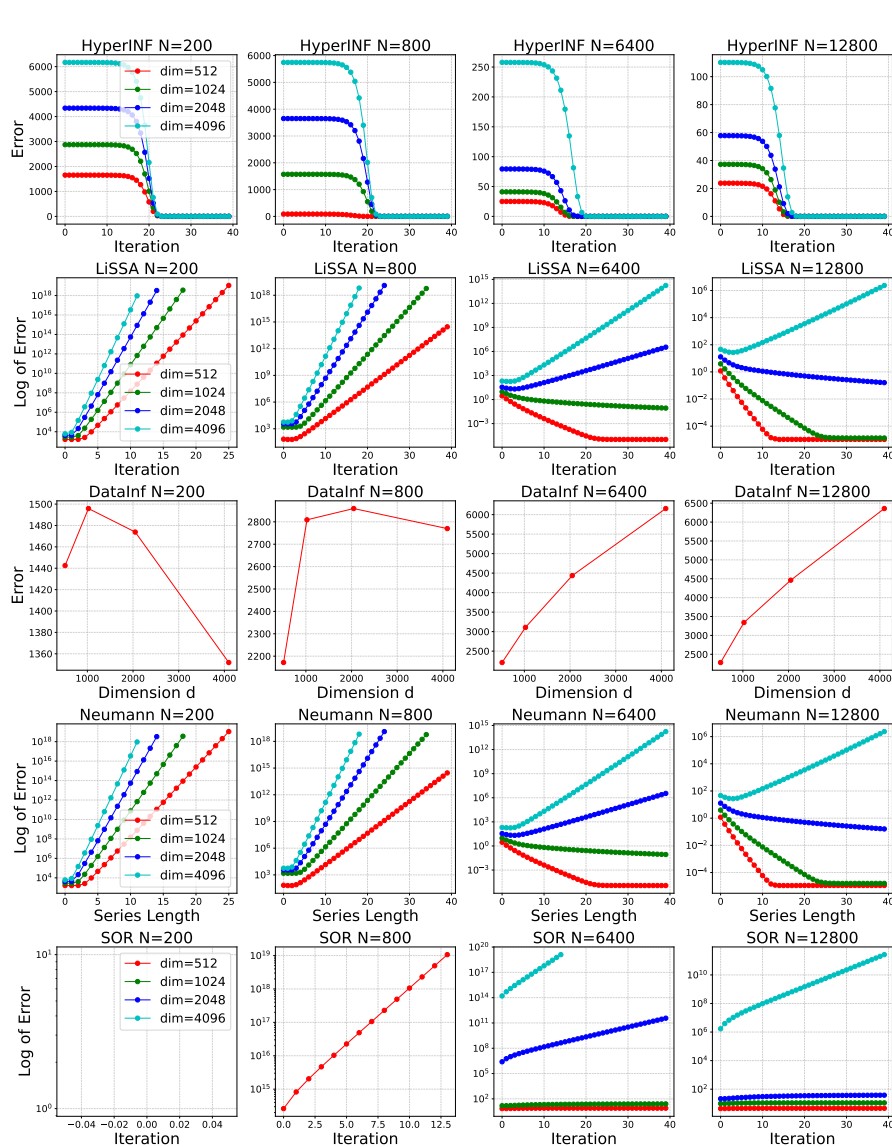

Figure 12: **Convergence test of HYPERINF, LISSA, DATAINF, Neumann Series, and SOR** ($\omega = 0.5$). We construct $M = \frac{1}{N}\sum_{i=1}^{N} s_i s_i^{\top} + \lambda I$ and apply various methods to approximate the inverse Hessian-vector product $M^{-1}v$, , where $s_i \in \mathbb{R}^d, v \in \mathbb{R}^d$ are randomly generated, each element is from the Standard Normal Distribution $\mathcal{N}(0,1)$. Only HYPERINF can converge to a low error rate in all cases. For LISSA, it does converge in some cases (e.g. $N = 6400, dim = 512$), but would diverge when $dim$ is larger. SOR only converges when $N$ is large and $dim$ is small.

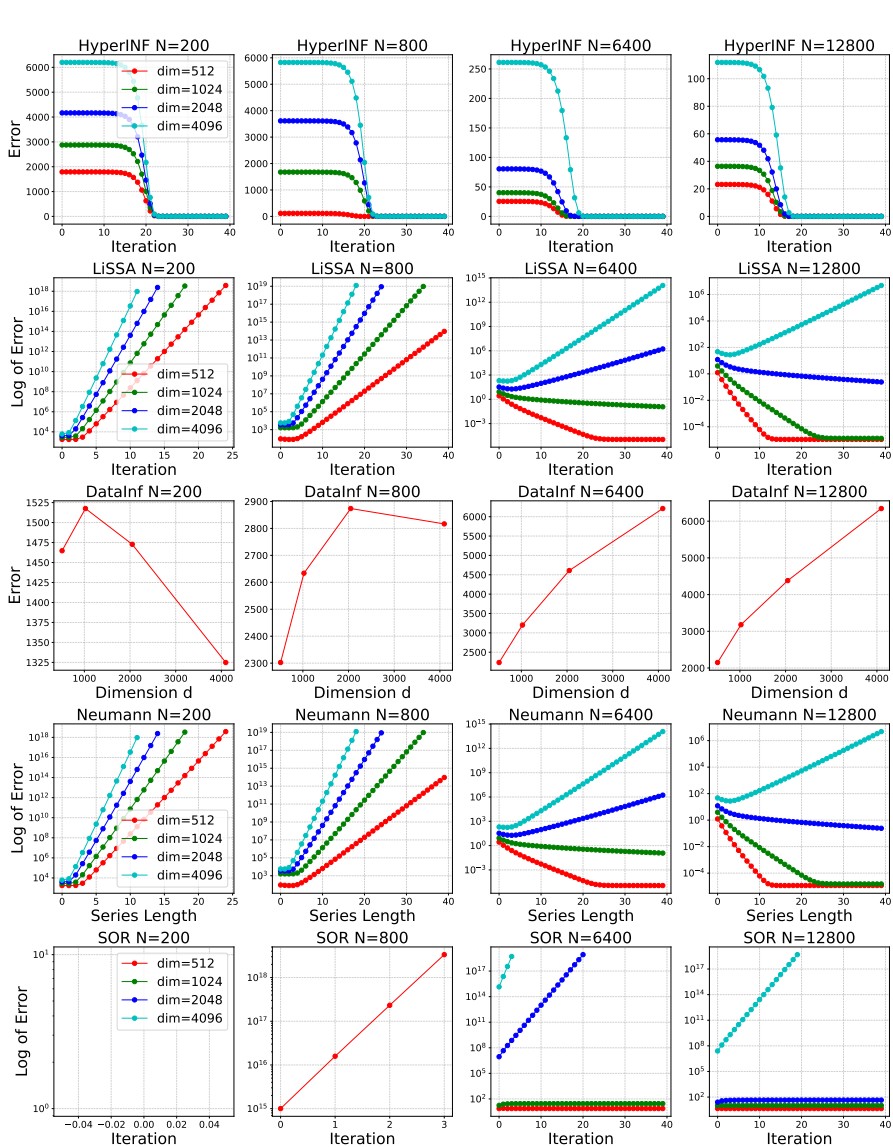

Figure 13: **Convergence test of HYPERINF, LISSA, DATAINF, Neumann Series, and SOR** ($\omega = 1.5$). We construct $M = \frac{1}{N}\sum_{i=1}^{N} s_i s_i^\top + \lambda I$ and apply various methods to approximate the inverse Hessian-vector product $M^{-1}v$, , where $s_i \in \mathbb{R}^d, v \in \mathbb{R}^d$ are randomly generated, each element is from the Standard Normal Distribution $\mathcal{N}(0, 1)$. Only HYPERINF can converge to a low error rate in all cases. For LISSA, it does converge in some cases (e.g. $N = 6400, dim = 512$), but would diverge when $dim$ is larger. SOR only converges when $N$ is large and $dim$ is small.

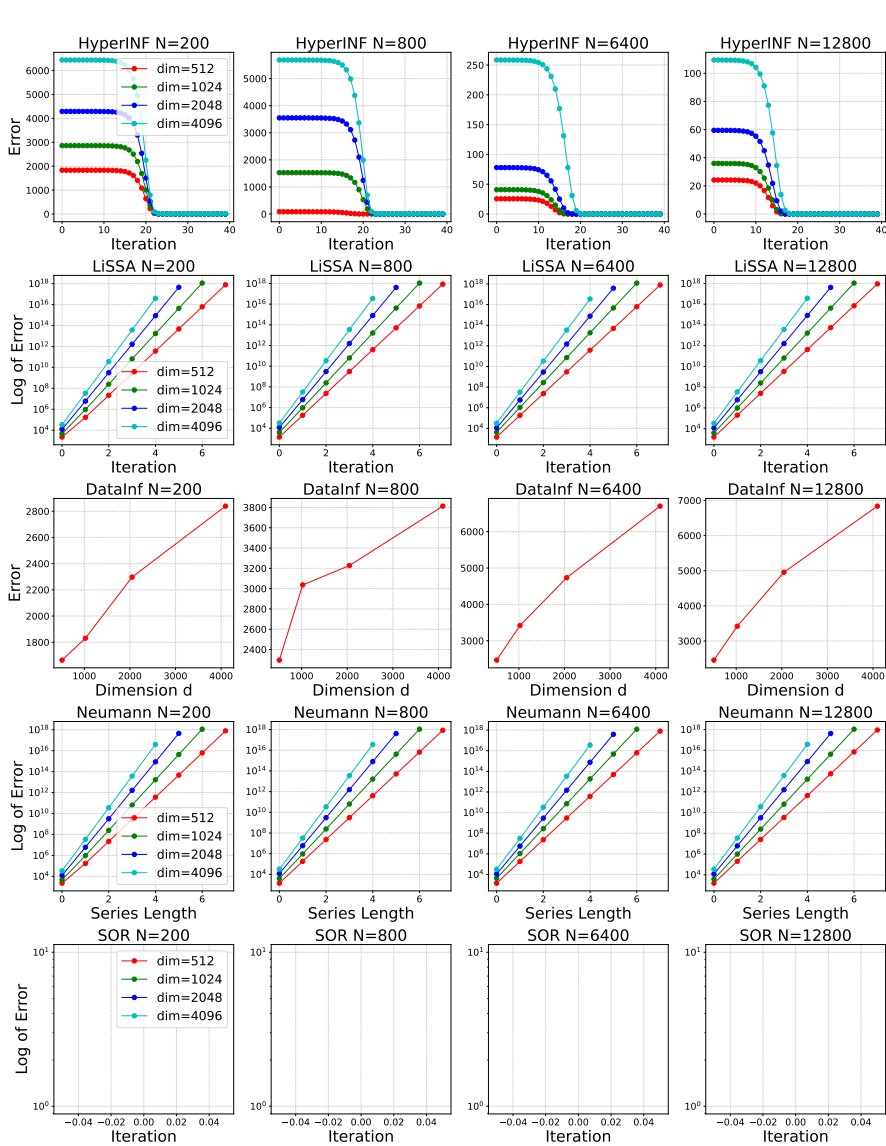

Figure 14: **Convergence test of HYPERINF, LISSA, DATAINF, Neumann Series, and SOR ($\omega = 1.5$).** We construct $M = \frac{1}{N}\sum_{i=1}^{N} s_i s_i^\top + \lambda I$ and apply various methods to approximate the inverse Hessian-vector product $M^{-1}v$, , where $s_i \in \mathbb{R}^d$, $v \in \mathbb{R}^d$ are randomly generated, each element is from th Normal Distribution $\mathcal{N}(0.5, 1)$. Only HYPERINF can converge to a low error rate in all cases. For other methods, they all diverge. For SOR, it has the `nan` issue.

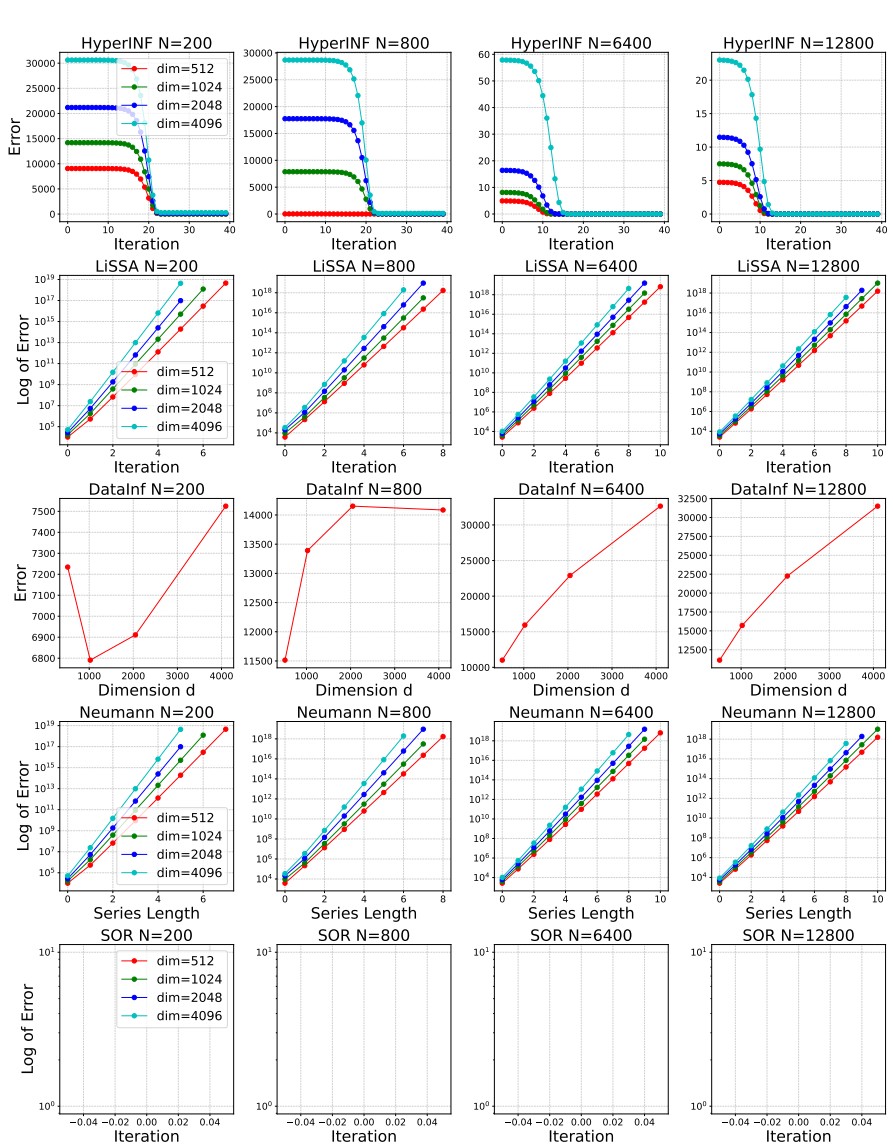

Figure 15: **Convergence test of HYPERINF, LISSA, DATAINF, Neumann Series, and SOR** ($\omega = 1.5$). We construct $M = \frac{1}{N} \sum_{i=1}^{N} s_i s_i^\top + \lambda I$ and apply various methods to approximate the inverse Hessian-vector product $M^{-1} v$, , where $s_i \in \mathbb{R}^d, v \in \mathbb{R}^d$ are randomly generated, each element is from the Normal Distribution $\mathcal{N}(0, 5)$. Only HYPERINF can converge to a low error rate in all cases. For other methods, they all diverge. For SOR, it has the `nan` issue.

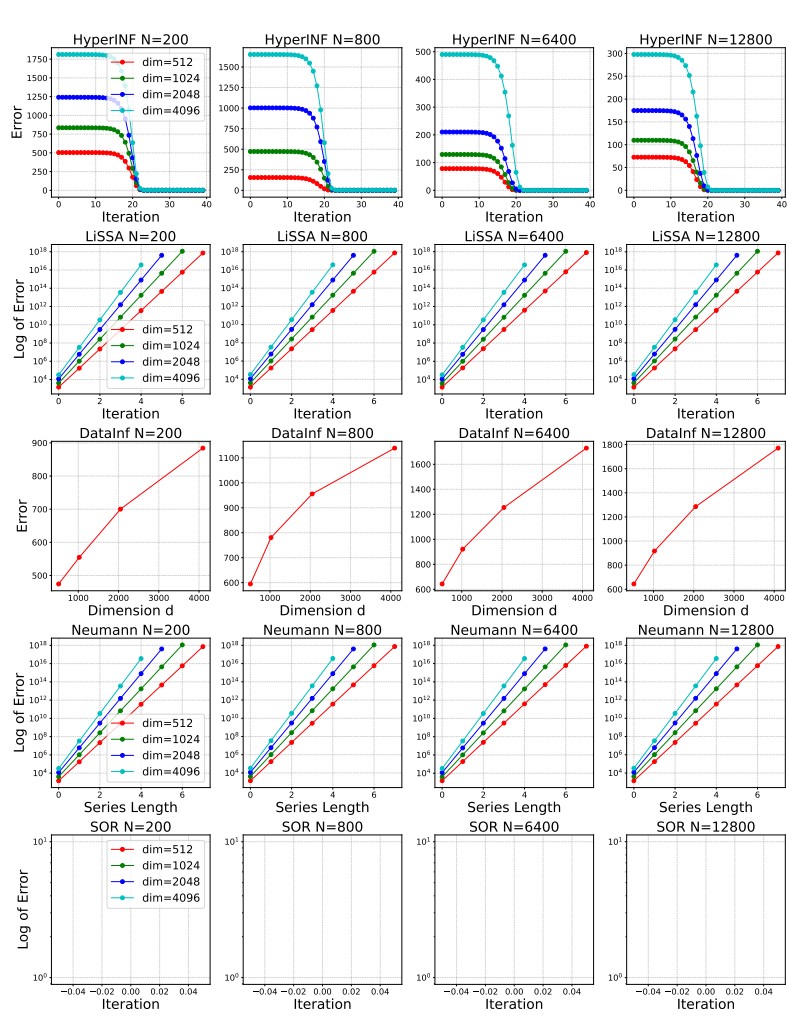

Figure 16: **Convergence test of HYPERINF, LISSA, DATAINF, Neumann Series, and SOR** ($\omega = 1.5$). We construct $M = \frac{1}{N} \sum_{i=1}^{N} s_i s_i^\top + \lambda I$ and apply various methods to approximate the inverse Hessian-vector product $M^{-1} v$, , where $s_i \in \mathbb{R}^d, v \in \mathbb{R}^d$ are randomly generated, each element is from the Uniform Distribution $U(0, 1)$. Only HYPERINF can converge to a low error rate in all cases. For other methods, they all diverge. For SOR, it has the `nan` issue.

# I DISCUSSION AND LIMITATIONS ON FIM AND GFIM APPROXIMATION IN INFLUENCE FUNCTION COMPUTATION

## I.1 LIMITATIONS OF FIM APPROXIMATION OF HESSIAN MATRIX

While the Fisher Information Matrix (FIM) have been widely applied to approximate the Hessian matrix (Bartlett, 1953; Kwon et al., 2024), we recognize that some infeasible conditions required by Equation 3 cannot be met in realistic LLM training cases, which might cause discrepancies and undesirable downstream effects. Firstly, Equation 3 only stands when the model is nearly converged, which can hardly be achieved when train LLMs; Besides, Equation 3 requires that the labels $y$ are drawn from the distribution $p(y|x, \boldsymbol{\theta})$. While the ground-truth labels are normally used as $y$ in influence function computation.

From the optimization point of view, using FIM to approximate second-order gradients or curvature during training could lead to sub-optimal optimization outcomes, wuch as adverse distortion of the gradient field (Kunstner et al., 2020). For more detailed and complete studies of FIM and hessian matrices, we refer the readers to (Kunstner et al., 2020).

## I.2 LIMITATIONS OF GFIM APPROXIMATION OF FIM

In Theorem 3.1, we make the idealized assumption that each column in the gradient matrix $g$ is independently and identically distributed (i.i.d.) following a distribution with zero-mean. However, we demonstrate that this assumption may not be strictly valid in realistic cases of large langauge model training.

According to 17a, we visualize both the fisher information matrix (FIM, $vec(g)vec(g)^T$) and expended generalized fisher information matrix (GFIM, $I_r \otimes gg^T$) of gradient matrices from LoRA finetuning on the MRPC dataset.

In 17b, we constructed a $16 \times 1000$ matrix by sampling each column from a standard guassian distribution with zero-mean and one-variance independently and identically. We then plot the FIM and expended GFIM matrices of the given matrix.

In practice, FIM and GFIM show some differences, especially with randomness and complex dynamics during LLM training. However, it does not impact the empirical performance of our method according to the improvement from our comprehensive experiments. How to derive a more accurate low-rank approximation of Hessian matrices within tractable computations is an important and compelling research topic. We will leave it for future work.

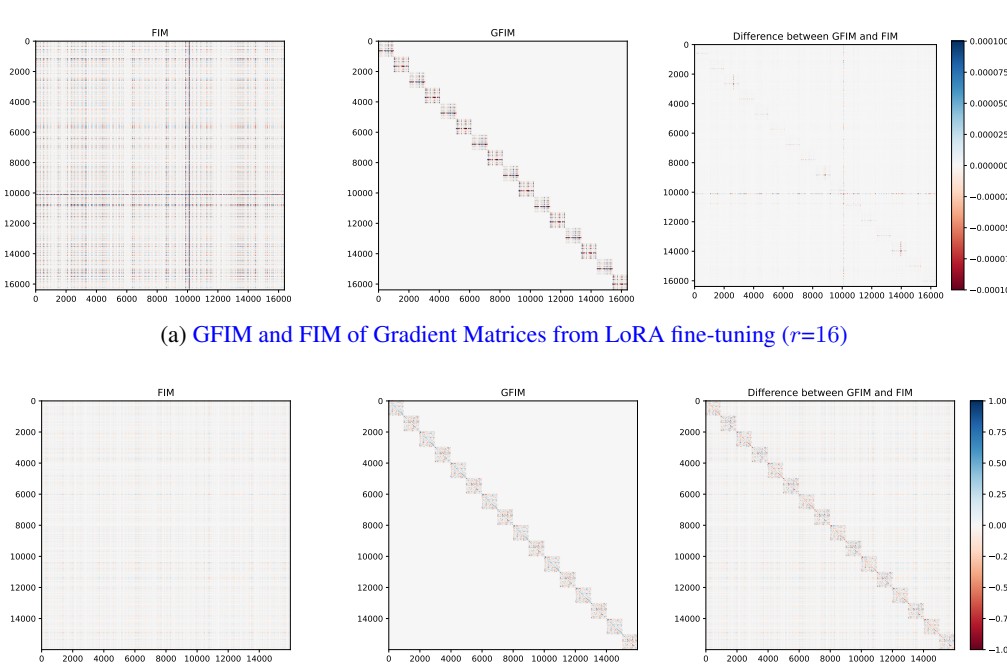

(a) GFIM and FIM of Gradient Matrices from LoRA fine-tuning ($r$=16)

(b) GFIM and FIM of Matrices sampled from Standard Gaussian Distribution

Figure 17: Difference between GFIM and FIM.

### I.3 LINEAR INDEPENDENCE OF MATRIX COLUMNS

In realistic LLM training, it is hard to justify the i.i.d. assumption made in Theorem 3.1. However, we provide the empirical evidence that each column in the gradient matrices are linear independent with each other. Specifically, the rank of the gradient matrix should be equal to the number of columns, i.e. the LoRA rank in low-rank fine-tuning.

We hereby compute the rank of each gradient matrix across all training data points from MRPC dataset and present the distribution of matrices ranks in 18a and 18b. With $r=8$ and $r=16$, most of ($> 90\%$) gradient matrices are with full column ranks, which shows that Theorem 3.1 stands in real low-rank tuning cases. In addition, we also compute the difference between GFIM and FIM in the above same setting ($r = 16$ in this experiment).

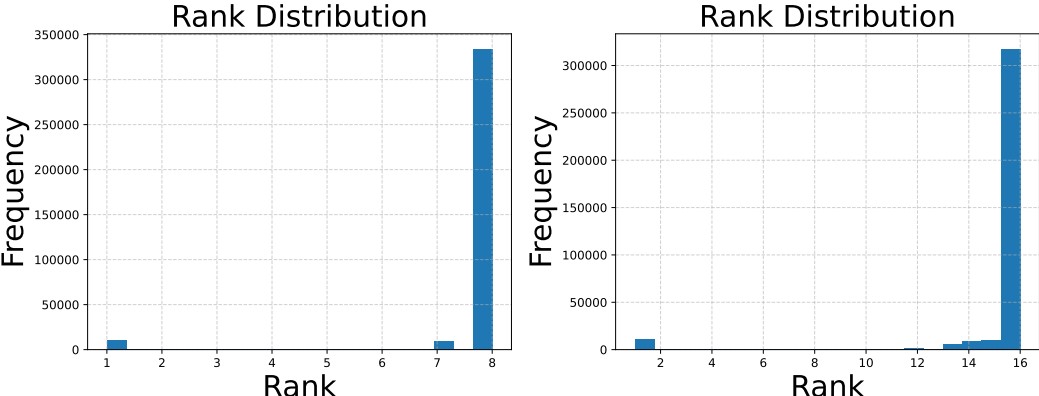

(a) Rank Distribution of Gradient Matrices with $r$=8.  (b) Rank Distribution of Gradient Matrices with $r$=16.

Figure 18: Rank distribution of gradient matrices on MRPC. More than $90\%$ matrices are with full column rank, which justifies our linear dependent aussumption in Theorem 3.1.

