# OpenReview forum: "HyperINF: Unleashing the HyperPower of the Schulz's Method for Data Influence Estimation"
_ICLR.cc/2025/Conference — Submitted to ICLR 2025_

### Official Review · Reviewer_3Qw4 · 2024-11-04

**Soundness:** 3
**Presentation:** 2
**Contribution:** 2
**Rating:** 5
**Confidence:** 3

**Summary:**

This paper investigates the problem of effective and efficient approximations for the Hessian matrix in Influence Functions for large-scale ML models. The paper points out that existing methods for this tasks, LiSSA and DataInf, have higher computation complexities and/or looser error bounds. This paper proposes HyperInf, which performs block-wise diagonal approximations of the Hessian matrix and its inverse and then approximate the inverse matrix with the iterative Schultz method. The paper shows that, empirically, this iterative Schultz method is robust to different initializations and, theoretically, enjoys a favorable convergence rate. The paper conducts empirical studies on mislabeled dection on the GLUE benchmark with BERT models, data selection for LLM fine-tuning, and training VLMs on instruction datasets and compares the performance of HyperInf with Random, TracIn, DataInf and LiSSA. Empirical results shows HyperInf achieves better overall performance.

**Strengths:**

The scope and motivation of the paper is clear. It is straightforward to understand what does this paper aim to improve.

The logical flow for theoretical development is coherent. The derivations are clear and the elobrations are accurate.

Experiments are diverse and the comparisons are comprehensive.

**Weaknesses:**

The work is heavily inspired by DataInf where many components are shared, such as the use of Fisher Information Matrix (FIM). Some theoretical analysis directly cites DataInf for results. This may pose challenges for some readers and mandate reading DataInf to fully understand this paper. It may be helpful to add additional introductions and comments of DataInf directly into the narrative of this manuscript and make it more indepedent.

**In general, the contribution of this paper appears incremental compared to DataInf. The major change is replacing the Sherman-Morrison formula to the iterative Schultz method for approximating matrix inversion.** Sherman-Morrison formula appears to be the standard approach for approximating matrix inversion, which is expected to work especially for low-rank matrices. This paper only cites its looser theoretical guarantee to motivate the development of the proposed method. In machine learning, theoretical bounds are often conservative which may or may not be relevant to the actual use case. There appears to be a major gap in the narrative and ncessciates in-depth comparisons and discussions, both theoreitcal and empirical.

The set of empirical studies is less conventional. Data selection for foundation models is known to be a tricky task where the selection scale has challenges the prior knowledge for the tradeoff between data quality and diversity. Influence-based method such as [Less: Selecting influential data for targeted instruction tuning] turns out to be less effective for selecting pre-training data and many selection methods may not outperform random baselines (Ref: [Rethinking Data Selection at Scale: Random Selection is Almost All You Need] ). Thus, I am not fully convinced by results on these experiments.

**Questions:**

Since these techniques are all designed for approximating the inverse of Hessian matrix, which is also a proxy for the difference in model performance compared to re-training the model without the sample.

Why not starting from fundamentals and conducting apple-to-apple comparisons on how each of these method approximate the inverse of Hessian matrix and how they relate to the actual leave-one-out error?

Besides, this paper misses the comparison for the actual compute overhead of each method. This could be a result of great interest.

---

> ### Author Response · Authors · 2024-11-28
> **Author response to reviewer 3Qw4**
>
> We thank the reviewer for the valuable feedback! We have updated the manuscript according to the suggestions, and provided further clarifications as follows:
> ## W1: Clarification of the main contributions
> The first contribution of this work is leveraging Generalized Fisher Information (GFIM, Equ. 4) in the data attribution problem, which yields a novel low-rank formulation of the data influence estimation function (Equ. 5). Secondly, we apply Schulz’s method (Equ. 6) to improve the accuracy of matrix inverse approximation. The mathematical formulas of Schulz’s method are from very old linear algebra literature, while it has not been used in large-scale neural network training or data attribution problems. By combining the two techniques, we can achieve improved efficiency and accuracy, which then transfer to better empirical performance. In comparison, the prior work (DataInf, [1]) only proposed to apply sherman-morrison formula upon previously proposed influence functions.
>
> ## W2: Convergence analysis on Schulz’s method
> We refer the reviewer to **Appendix C**, where we include a proof of convergence of Schulz’s method. Indeed, we agree with the reviewer that not all theoretical improvements can be demonstrated useful empirically. However, we show with comprehensive empirical results that our proposed method outperforms all the other baselines in various real-world data attribution tasks.
>
> ## W3: Balance of Quality and Diversity
> Firstly, we want to clarify that HyperINF is proposed to improve the accuracy and efficiency of the influence function computation on large-scale models. We demonstrate from comprehensive experiments that our method brings a large improvement in various real-world tasks. The discussion on the balance of quality and diversity may be beyond this scope.
> Also, our work mainly focuses on task-specific fine-tuning or instruction tuning instead of multi-task learning or general-purposed training. In that case, we claim the relevance to the target task and the quality of data points would indeed have a great impact on the downstream performance. Regarding the criticism of [2] and the results from [3], one potential explanation is the whole training data distribution is too far from the target task, which makes random selection a very strong baseline. Besides, [3] did not report the selection ratios in their paper, which makes their results less convincing.
>
> We consider the study on the balance of data quality and diversity can be an intriguing topic, especially on multi-task learning. We will leave it for future work.
>
> ## Q1: Convergence experiments among different methods and Leave-One-Group-Out error experiments
> We conduct the correlation test with Leave-One-Group-Out (LOGO) retraining scores and present the results in **Appendix D.2**. To prevent the large computations incurred from retraining LLMs, we retrain the model by excluding each group of data (Leave-One-Group-Out) instead of each individual datapoint (LOO). the details can be found in Appendix D.2. The results demonstrate greatest correlation with the LOGO, which indicates its superior accuracy on data attribution.
>
> ## Q2: Running time comparisons
> We refer the reviewer to **Appendix D.1**, where we provide both the time costs on CPU and GPU across various LoRA ranks. We observe that the efficiency of three algorithms ranks largely differently between GPU and CPU. On CPU (Figure 6), DataInf introduces the least time overheads while HyperINF incurs the most amount of extra time costs. In addition, the time costs from DataInf and LiSSA increase quadratically with LoRA rank $r$ while HyperINF increases linearly. Alternatively, on one single GPU (Figure 7), the time costs from all algorithms are almost constant across LORA ranks, and HyperINF costs the least amount of time, followed by DataInf. In comparison, LISSA requires ~4x more time costs than HyperINF and DataInf. **That difference demonstrates that HyperINF is more efficient and compatible with modern GPU computing.**
>
> We hope our answers resolve most of your concerns. If you have any further questions, don’t hesitate to contact us. Your valuable suggestions have helped us improve the manuscript!
>
> [1] DataInf: Efficiently Estimating Data Influence in LoRA-tuned LLMs and Diffusion Models
>
> [2] Less: Selecting influential data for targeted instruction tuning
>
> [3] Rethinking Data Selection at Scale: Random Selection is Almost All You Need

---

> > ### Comment · Reviewer_3Qw4 · 2024-12-01
> >
> > Thanks for the response. I appreciate the authors' effort in improving this work. I read through other reviews, authors' responses, and revisions in the manuscript, etc. I also noticed the discussion with Reviewer mprn on the convergence properties of LiSSA and its implementations.
> >
> > **I have a hard time giving a single rating to this manuscript.**
> >
> > I'm not fully convinced of the conceptual contribution of this work. Theoretically, I still don't find replacing Sherman-Morrison formula with Schulz’s method for approximating inverse Hessian particularly novel. But I also think judging the theoretical improvements with a simple score might be inappropriate.
> >
> > Empirical evidence is relatively weak. These experiment setups are not directly relevant to practical use cases. The results appear highly noisy as the baseline methods underperform random in half of the cases. It is hard to gauge the usefulness of the proposed method with these results.
> >
> > I changed my rating to 5 and reduced my confidence to 3.

---

> > > ### Author Response · Authors · 2024-12-02
> > >
> > > We thank the reviewer for adjusting the score! We have reimplemented LiSSA according to reviewer mrpn's suggestion and present the results in the anonymous codebase (https://anonymous.4open.science/r/HyperINF-B702/rebuttal/hvp_converge_correct_lissa.pdf). The result demonstrates that HyperINF still consistently outperform LiSSA after the correction.
> > >
> > > We also thank the reviewer for the thoughtful consideration in the empirical results! We will run more experiments in various settings (e.g. with different level of noises) and improve our final manuscript according to the suggestions!
> > >
> > > Sincerely,
> > >
> > > the author

---

### Official Review · Reviewer_7qzk · 2024-11-04

**Soundness:** 3
**Presentation:** 3
**Contribution:** 3
**Rating:** 5
**Confidence:** 3

**Summary:**

This paper addresses an important challenge in scaling influence functions to large-scale models by introducing HYPERINF. The authors propose combining Schulz's method with a generalized Fisher Information Matrix, achieving improved convergence performance for Hessian matrix inversion with greater efficiency. Through comprehensive experiments across mislabeled data detection and data selection tasks for both LLMs and VLMs, the method shows significant advantages over existing baselines.

**Strengths:**

1. The authors tackle the important challenge of estimating data influence in large-scale models.
2. The authors propose leveraging the GFIM to enhance computational efficiency and Schulz's method to improve the convergence guarantee, presenting a technically intriguing approach. They further demonstrate superior convergence performance over baselines in controlled settings.
3. The authors provide extensive experimental validation to support the effectiveness of their proposed method.

**Weaknesses:**

Please see the Questions.

**Questions:**

1. Lemma 1 relies on the assumption that gradient columns are independently and identically distributed. However, it's unclear what this means in practice. For a gradient matrix, how can we justify that each column is independent? The authors should provide more insights or practical explanations for this assumption.
2. While the proposed method shows clear advantages in controlled settings, it sometimes underperforms compared to baselines in practical applications, as indicated in Tables 2 and 3. What insights do the authors have about this discrepancy? Is this due to violated assumptions, or might the high-influence points selected by the proposed method not necessarily translate to improved accuracy?
3. I'm more concerned about the practical applications of this approach in LLM data selection. For example, when evaluating training point influence, should we only focus on D_val from the same distribution as D_train? As a model developer, I would be more interested in understanding how incorporating certain training data would improve general performance rather than just performance on the same distribution. Specifically, 5% of data selected from the proposed method can improve the general performance if the D_val consists of samples from multiple domains.
4. Since the performance gap may vary depending on the dataset, could the authors explain their criteria for selecting evaluation datasets?
5. Do the authors could further explain why dense fine-tuning shows strong performance with 5% data selection, compared to Lora-finetuned models?

---

> ### Author Response · Authors · 2024-11-28
> **Author response to reviewer 7qzk**
>
> We thank the reviewer for the valuable feedback! We have updated the manuscript according to the suggestions, and provided further clarifications as follows:
>
> ## Q1: Empirical evidence justification for Lemma 1
> We agree with the reviewers that our assumptions made for Lemma 3.1 are idealized and hard to verify in real large language model training. Instead, we want to demonstrate that in some conditions, the generalized fisher information matrix (GFIM) and fisher information matrix (FIM) are equal in expectation, which motivates our low-rank formulation of the information function (Equ. 5).
> We also thank the reviewer for the great references! We include a detailed discussion on the limitations of GFIM and FIM approximations in Appendix I. In practice, FIM and GFIM could have some differences (Figure 17), while it does not impact the empirical performance of our method according to the improvement from our comprehensive experiments.
> How to derive a more accurate low-rank approximation of Hessian matrices within tractable computations is an important and compelling research topic. We will leave it for future work.
>
> ## Q2: Potential explanation for the strong performance by random baseline
> We agree with the reviewer that the random sampling baseline demonstrates strong performance in some cases but we want to clarify that HyperINF could outperform it in most of the tasks. We provide potential explanations as follows:
> (1). Since all the influence-function-based methods could ignore the interactions between various data points, which could suffer from duplicated data points and lead to suboptimal performance from the lack of diversity.
> (2). The training dataset could already include relevant and high-quality data points. Therefore, the data selection may not bring much improvement above random sampling (e.g. Hellaswag).
>
> ## Q3: Practical applications concerns for multi-task
> We want to clarify that our work mainly focuses on task-specific fine-tuning or instruction tuning instead of multi-task learning or general-purposed training. Thus, we only conducted data selection experiments on LLM finetuning with one specific target domain, where the $D_{train}$ and $D_{val}$ are from the same distribution. The multi-task data attribution problem is a more complex setting, where each target could either overlap or conflict with other ones. We consider it as a very intriguing direction for data attribution and we will leave it to future work.
>
> ## Q4: Evaluation datasets selection
> We provide some insights from our data selection experiments. Firstly, we find the influence-function-based methods could bring more improvement on more challenging tasks (e.g. QASC, LogiQA) while having marginal improvements on easier ones (e.g. Hellaswag). Another insight is that the data attribution method would be less effective when all the training data points are very high-quality or the whole training dataset is far from the target task.
>
> On the split of validation and training data, we randomly sample $D_{val}$ from the same distribution of $D_{train}$, without applying quality control. However, we expect using a well-curated high-quality subset as the validation set ($D_{val}$) could lead to better data attribution performance.
>
> ## Q5: Strong performance on dense finetuning than LoRA-finetuning
> We thank the reviewer for the detailed and interesting observation! We provide a potential explanation as follows: since we use the dense gradient to estimate the influence function, we expect HyperINF could yield more accurate scores, while other baseline methods could suffer from large approximation errors. Thus, it will select more relevant and high-quality datapoints for finetuning, which makes the difference more distinguishable, especially when selecting a small portion of data points (5%).
>
> We hope our answers resolve most of your concerns. If you have any further questions, don’t hesitate to contact us. Your valuable suggestions have helped us improve the manuscript!

---

> ### Author Response · Authors · 2024-12-02
> **End of the discussion period**
>
> Dear reviewer 7qzk,
>
> Thank you again for your efforts reviewing our paper and providing valuable feedback! Since **tomorrow is the last day of the author-reviewer discussion phase**, we hope you can take some time to look into our rebuttals and additional results. Let us know if you have other concerns and questions!
>
> Sincerely,
>
> the authors

---

> > ### Comment · Reviewer_7qzk · 2024-12-03
> > **Response to Authors' Rebuttal**
> >
> > Thank you for your rebuttal and clarification. I appreciate the authors' effort in preparing the rebuttal and improving the manuscript based on the reviews provided.
> >
> > It is surprising that other baselines (e.g., DATAINF, LISSA, TracIN) sometimes show even lower performance than random selection (e.g., 5% AVG DATAINF, LISSA, TRACIN in Table 8, 5% AVG DataINF in Table 3, 5% and 20% AVG LISSA in Table 4). Additionally, the results vary across different selection ratios (e.g., in Table 2, QASC random selection shows the best performance at 5%, while influence-based methods show better performance at 20%), which is interesting to analyze, but at the same time, it makes me difficult to understand why each method shows different advantages across data distributions and ratios without enough discussion. While the authors explain this in terms of influence-based methods, the results differ among these methods (e.g., sometimes DataINF shows the best results, and sometimes LISSA performs best). It would be helpful if the authors could provide more discussion on this. Additionally, I asked about the rationale of the current evaluation dataset selection. Although the authors provided insights about which datasets are suitable for influence-based methods, it does not answer the point of the selection of four datasets in Tables 2 and 3. For example, is this experiment designed for the reasoning task? will the result be consistent even if we consider a dataset from a different task? The reason for this question is the gap between methods looks marginal and might be flipped if we consider different datasets (e.g., Table 2, 0.2% gap for 5% selection, and 1% for 20% selection, Table 4, 0.2% for 5% selection, 1.6% for 20% selection). Thus, the consistency of these results across different datasets, and models remains a concern.
> >
> > On the other hand, this method works well on the mislabeled data detection task, except for SST2. It would be also helpful if the authors could discuss this in more detail.

---

### Official Review · Reviewer_GxRJ · 2024-11-04

**Soundness:** 2
**Presentation:** 3
**Contribution:** 2
**Rating:** 5
**Confidence:** 4

**Summary:**

This paper is motivated by the computational challenge of influence functions, and proposes an approximation method based on the Schulz's iterative algorithm. In particular, the generalized fisher information is used to construct a low-rank approximation of Hessian matrix. Experiments of proposed method is illustrated on simulation, mislabeled data detection,  data selection for LLM, and VLM fine-tuning.

**Strengths:**

- This paper proposes a principled method to approximate influence function, which is a very important topic in modern learning with large-scale models and datasets
- The proposed method is evaluated on various applications including LLM training

**Weaknesses:**

Major
- I was put off when noticing that the general setup seems to include a general loss function $\ell$, yet the derivation relies on Equation (3), which restricts $\ell$ to be a log-likelihood function.
- From Table 1, it is clear that the proposed method, HyperINF, performs worse than the existing method, DataInf, in terms of all three complexities. The paper claims that DataInf has a $O(d^2)$ approximation error, making it prone to large approximation errors when $d$ is large. However, there is a lack of rigorous characterization or illustration of HyperINF's performance concerning approximation error.
- In particular, I am surprised to see that, even though the abstract mentions leveraging the low-rank structure, the proposed algorithm’s computational complexity barely depends on $r$, with even less dependence than DataInf.
- Regarding estimation accuracies in Table 2, the performance difference compared to DataInf is marginal or even worse in some cases.

Minor
- Lemma 1 is essentially the same as Lemma 1 in the cited reference Yang et al. (2022). I think the proof is unnecessary, and credit should be given to the reference in the main text of the paper, rather than just saying "following the proof of Yang et al. (2022)" in the appendix.
- Equation (3): I believe a negative sign is missing.
- Figure 2: The plots are not color-blind-friendly.

**Questions:**

- Overall, I am not fully convinced that the proposed method is broadly applicable, and its theoretical contribution feels rather thin. I would appreciate it if the authors could clarify the mathematical derivations of the proposed method and address the weaknesses mentioned above.
- Regarding the numerical experiments, could the authors consider building additional experimental setups or metrics that go beyond accuracy comparisons, such as directly evaluating convergence speed, running time, etc.? This would also help present a more coherent narrative on the strengths of the proposed method.

---

> ### Author Response · Authors · 2024-11-28
> **Author response to reviewer GxRJ**
>
> We thank the reviewer for the valuable feedback! We have updated the manuscript according to the suggestions, and provided further clarifications as follows:
>
> ## W1: Choice of objective function
> We agree with the reviewer that the GFIM and FIM approximation of the Hessian matrix is only applicable when the loss function is a log-likelihood function, which is the mainstream objective function for auto-regressive model training. Given that a large portion of foundation models are trained in an auto-regressive way, our method can be widely applied to a wide range of tasks and models.
>
> ## W2&Q1: Theoretical convergence analysis on Schulz’s method
> We thank the reviewer for the suggestion on a more rigorous theoretical analysis of our method! We provide a detailed convergence analysis of Schulz’s method in **Appendix C**, where we show the approximation error decreases quadratically.
>
> ## W3: Analysis of Computing Time Costs
> We thank the reviewer for this intriguing observation! We have also discovered this discrepancy between theoretical complexity and real GPU hours, and we are happy that we may have found the answer. According to **Appendix D.1**, we provide both the time costs on CPU and GPU across various LoRA ranks. We observe that the efficiency of three algorithms ranks largely differently between GPU and CPU. **On CPU (Figure 6), DataInf introduces the least time overheads while HyperINF incurs the most amount of extra time costs. In addition, the time costs from DataInf and LiSSA increase quadratically with LoRA rank $r$ while HyperINF increases linearly. Alternatively, on one single GPU (Figure 7), the time costs from all algorithms are almost constant across LORA ranks, and HyperINF costs the least amount of time, followed by DataInf. In comparison, LISSA requires ~4x more time costs than HyperINF and DataInf.** That difference demonstrates that HyperINF is more efficient and compatible with modern GPU computing.
>
> ## Q2: Convergence test among different methods
> For the evaluation on convergence ability and speed, we refer the reviewer to Section 4 and Figure 1, where we compare the approximation error of the inverse Hessian-vector product from three different algorithms. We further conducted a complete study with matrices drawn from different distributions in Appendix H. The results demonstrate that HyperINF, based on Schulz’s method, consistently shows superior stability and convergence performance, which is robust with various data distributions. In comparison, the other methods can only converge in a few cases, indicating their sensitivity to different data distributions and the scale of matrices.
>
> ## W4: Marginal improvement than DataInf
>  The approximation error of DataInf algorithms is $O(d^2)$. In LoRA finetuning (Table 2&4), the number of tunable parameters per layer is relatively small, whereas DataInf could yield a small approximation error which has less impact on the rank of data samples. In the dense finetuning case (Table 3), HyperINF could have a larger improvement above DataInf when we have a larger amount of tunable parameters.
>
> ## W6: Missed negative sign in Equ. 3
> In our settings, we consider the loss function $l(\cdot)$ to be a **negative** log-likelihood function, rather than a log-likelihood function. Therefore, we believe Equ. 3 should be valid without a negative sign.
>
> ## W7: Color-blind plot
> We thank the reviewer for the considerate suggestion! However, it is a bit hard to make Figure 2 color-blind-friendly. In that case, we have included a table (Table 5) with detailed accuracies in **Appendix D**.
>
> We hope our answers resolve most of your concerns. If you have any further questions, don’t hesitate to contact us. Your valuable suggestions have helped us improve the manuscript!

---

> ### Author Response · Authors · 2024-12-02
> **End of the discussion period**
>
> Dear reviewer GxRJ,
>
> Thank you again for your efforts reviewing our paper and providing valuable feedback! Since **tomorrow is the last day of the author-reviewer discussion phase**, we hope you can take some time to look into our rebuttals and additional results. Let us know if you have other concerns and questions!
>
> Sincerely,
>
> the authors

---

### Official Review · Reviewer_v6An · 2024-11-04

**Soundness:** 3
**Presentation:** 3
**Contribution:** 2
**Rating:** 5
**Confidence:** 3

**Summary:**

This paper proposes a new influence function approximation scheme (HYPERINF) based on Schulz’s iterative algorithm for matrix inversion and the generalized Fisher Information Matrix. The authors elaborate on the intuitions and the details of the algorithm, and also numerical experiments demonstrate the algorithm's advantages over benchmark methods.

**Strengths:**

The paper is well-written and easy to follow. It provides a thorough account of the existing methods for the problem. The proposed method is new and naturally integrates existing ideas for approximating the Hessian matrix. Numerical results show the promise of the method compared to benchmarks.

**Weaknesses:**

I have the following comments about the paper:

My main concern is that the proposed method is a combination of several existing ideas:
- The general Fisher information matrix is from the DataInf paper of Kwon et al. (2024)
- The blockwise structure of the Hessian matrix is from Zhang et al. (2024) a;b
- The Schultz’s method is a well-known algorithm for matrix inversion.
To this end, the proposed method is more like an ad-hoc engineering improvement of the existing method.

More specifically, if we view it as an extension of Kwon et al. (2024) that adopts the Schultz’s method, the computational advantage and accuracy improvement against DataInf in Kwon et al. (2024) seem not quite significant to me.

In the numerical experiment, it seems unfair to evaluate LISSA using the Frobenius norm when v is randomly generated. If LISSA performs poorly as the authors argue (e.g., with an approximation error around 10^5 as noted on line 301), why does LISSA outperform other methods in some cases, such as for HellaSwag on line 411?

Also, how are some of the experimental parameters chosen? For example, rank = 64 is used in some cases and rank = 16 in others, why this choice? Why is the comparison in line 388 limited to training the model on the full dataset for just one epoch?

On the theoretical side, the computational complexity of equation (23) should be O(d^2), while the complexity of equation (22) should be O(d^3). Even though these complexities may be further optimized, they can never be reduced to O(d) and O(d^2) as I understand.

Minor note:
1). Line 207, identify -> identity
2). In equation (4), there shouldn't be a factor 1/r in the last term.

**Questions:**

See above.

---

> ### Author Response · Authors · 2024-11-28
> **Author response to reviewer v6An**
>
> We thank the reviewer for the valuable feedback! We have updated the manuscript according to the suggestions, and provided further clarifications as follows:
>
> ## W1: Clarification of the main contributions
> The first contribution of this work is leveraging Generalized Fisher Information (GFIM, Equ. 4) in the data attribution problem, which yields a novel low-rank formulation of the data influence estimation function (Equ. 5). Secondly, we apply Schulz’s method (Equ. 6) to improve the accuracy of matrix inverse approximation. The mathematical formulas of Schulz’s method are from very old linear algebra literature, while it has not been used in large-scale neural network training or data attribution problems. By combining the two techniques, we can achieve improved efficiency and accuracy, which then transfer to better empirical performance. In comparison, the prior work (DataInf, [1]) only proposed to apply Sherman-Morrison formula to previously proposed influence functions. We consider our contribution is not incremental compared to DataInf.
>
> ## W2: Fair comparisons among different methods on convergence tests
> We thank the reviewer for the good advice! *We updated Figure 1* with the comparison of the inverse Hessian-vector product $H^{-1}v$ from all three methods. We also conducted a complete study with matrices drawn from different distributions in **Appendix H**. The results demonstrate that HyperINF, based on Schulz’s method, consistently shows superior stability and convergence performance, which is robust with various data distributions. In comparison, the other methods can only converge in a few cases, indicating their sensitivity to different data distributions and the scale of matrices.
>
> ## W3: Choices of LORA ranks and number of training epochs
> For LLM finetuning experiments, we use LORA rank $r$=$64$ since the variance in data quality could be more distinguishable if the model has more finetuning capacity.  In the mislabeled data detection task, **we have also presented the results with LORA rank $r$=$8$ and $64$ in Appendix D**. We refer the reviewer to Figure 3, 4, 5 for details.
>
> We train the model on the full dataset for 1 epoch since it costs the same amount of FLOPs as 5 epochs on 20% selected data points. We also clarify this in Line 398-399 that, with 5% datapoints, HyperINF can achieve a comparable performance as the full dataset with 20$\times$ less tokens and $4\times$ less FLOPs.
>
> ## W4: Computational complexity of DataINF
> Since Equ. 23 and 24 are from one of the baseline methods [1], we refer the reviewer to the detailed discussion in DataINF [1].
>
> We hope our answers resolve most of your concerns. If you have any further questions, don’t hesitate to contact us. Your valuable suggestions have helped us improve the manuscript!
>
> [1] DataInf: Efficiently Estimating Data Influence in LoRA-tuned LLMs and Diffusion Models

---

> ### Author Response · Authors · 2024-12-02
> **End of the discussion period**
>
> Dear reviewer v6An,
>
> Thank you again for your efforts reviewing our paper and providing valuable feedback! Since **tomorrow is the last day of the author-reviewer discussion phase**, we hope you can take some time to look into our rebuttals and additional results. Let us know if you have other concerns and questions!
>
> Sincerely,
>
> the authors

---

### Official Review · Reviewer_mprn · 2024-11-04

**Soundness:** 3
**Presentation:** 3
**Contribution:** 2
**Rating:** 5
**Confidence:** 4

**Summary:**

The paper proposes using the Schulz's method to approximate the inverse of Hessian matrices that appears in the influence function. The authors demonstrate in Figure 1 that the Schulz's method can converge to the ground-truths while other baselines, such as LiSSA, diverge. Besides, their experiments on downstream tasks show that better approximation of inverse could lead to better performance.

**Strengths:**

The experiment results strongly support the benefits of more accurately approximating the inverves of Hessian matrices when using influence function.

**Weaknesses:**

- I did not see any differences between Lemma 1 presented in the submission and Lemma 1 in (Yang et al., 2022).

- The paper seems to suggest that the performance boost is mainly due to a more accurate approximation of the inverse of Hessian matrices in the influence function. In Figure 1, it shows that LiSSA does not converge, which should not be the case. My guess is that the authors adopted the implementation from Koh and Liang (2017), which may not align well with the theory. That being said, LiSSA can be implemented in a way that guarantees convergence; see Appendix D in (Bae et al., 2022). Therefore, I do not think the authors have made a fair comparison in Figure 1.

Minor:

- Regarding the introduction of the influence function, I would suggest that the authors refer to (Bae et al., 2022) for how the ideal assumptions of strong convexity and attainable optimal solutions can be mitigated.

- I would recommend that, regarding Eq. (3), the authors explicitly write down that the two expectations are taken over $(X,Y) \sim p(X)p(Y|f\_{\boldsymbol\theta}(X))$ for clarity. It demonstrates that the approximation holds if $p(Y|f\_{\boldsymbol\theta}(X))$ fits the training data well. Nevertheless, in the context of Eq. (3), this approximation is always there by using the first-order Taylor approximation of $f\_{\boldsymbol\theta}(X)$.

Overall, I feel that this work's main contribution is demonstrating that better approximations of inverse, when using the influence function, may lead to improved downstream performance.

Bae, J., Ng, N., Lo, A., Ghassemi, M., & Grosse, R. B. (2022). If influence functions are the answer, then what is the question?. Advances in Neural Information Processing Systems, 35, 17953-17967.

Koh, P. W., & Liang, P. (2017). Understanding black-box predictions via influence functions. In International conference on machine learning (pp. 1885-1894). PMLR.

Yang, M., Xu, D., Cui, Q., Wen, Z., & Xu, P. (2022). An efficient Fisher matrix approximation method for large-scale neural network optimization. IEEE Transactions on Pattern Analysis and Machine Intelligence, 45(5), 5391-5403.

**Questions:**

Suppose that a Hessian matrix $H$ is invertible and satisfies $\\|H\\| \leq U$ for some scalar $U>0$, what are the complexities of using LiSSA and Schulz's method to approximate $H^{-1}$?

---

> ### Author Response · Authors · 2024-11-28
> **Author response to reviewer mprn**
>
> We thank the reviewer for the valuable feedbacks! We provide the further clarifications as follows:
>
> ## W1: Clarification on Lemma 3.1
> We indeed leverage the GFIM (i.e. MatFIM) following the same idea as Yang et al. [1], which makes our Lemma 3.1 similar to Lemma 1 in [1]. However, we further extend GFIM on large-scale data attribution and yield a low-rank influence function formulation (Equ. 5) in matrix form.
> We agree with the reviewers that our assumptions made for Lemma 3.1 are idealized and hard to verify in real large language model training. Instead, we want to demonstrate that in some conditions, the generalized fisher information matrix (GFIM) and fisher information matrix (FIM) are equal in expectation, which motivates our low-rank formulation of the information function (Equ. 5).
> We also thank the reviewer for the great references! We include a detailed discussion on the limitations of GFIM and FIM approximations in **Appendix I**.
>
> ## W2: Fair comparisons between LiSSA and others
> We indeed adopt the implementation of LiSSA from Koh and Liang (2017), could the reviewer clarify how to make the comparison more fair?
> For a fair comparison and complete study on the convergence test, we include the experiment results with matrices drawn from different distributions in **Appendix H**. The results demonstrate that HyperINF, based on Schulz’s method, consistently shows superior stability and convergence performance, which is robust with various data distributions. In comparison, the other methods can only converge in a few cases, indicating their sensitivity to different data distributions and the scale of matrices.
>
> ## W3: Clarification on our main contributions
> We thank the reviewer for recognizing the empirical improvement by applying our proposed method! We hereby want to clarify the main contribution of our work as two points: the first contribution of this work is leveraging Generalized Fisher Information (GFIM, Equ. 4) in the data attribution problem, which yields a novel low-rank formulation of the data influence estimation function (Equ. 5). Secondly, we apply Schulz’s method (Equ. 6) to improve the accuracy of matrix inverse approximation. While the mathematical formulas of Schulz’s method are from very old linear algebra literature, it has not been used in large-scale neural network training or data attribution problems. By combining the two techniques, we can achieve improved efficiency and accuracy, which then transfer to better empirical performance. In comparison, the prior work (DataInf [1]) only proposed to apply sherman-morrison formula upon previously proposed influence functions.
>
> ## Q1: Complexity of using LiSSA and Schulz’s method
> We compare the error on the inverse Hessian-Vector Product $H^{-1}v$ since LiSSA is not used to approximate $H^{-1}$ but directly estimate the inverse Hessian-Vector Product $H^{-1}v$.
> For the complexity comparison, we refer the reviewer to Table 1 for more details.
>
> We hope our answers resolve most of your concerns. If you have any further questions, don’t hesitate to contact us. Your valuable suggestions have helped us improve the manuscript!
>
> [1] An Efficient Fisher Matrix Approximation Method for Large-Scale Neural Network Optimization
>
> [2] DataInf: Efficiently Estimating Data Influence in LoRA-tuned LLMs and Diffusion Models

---

> ### Comment · Reviewer_mprn · 2024-11-30
> **Unfair and confusing convergence test**
>
> Thank you for your response.
>
> **My point is that LiSSA is guaranteed to converge in theory**. The issue that LiSSA diverges in Figure 1 originates from Koh and Liang (2017), whose interpretation of LiSSA differs from its original proposal by Agarwal et al. (2017). Suppose a Hessian matrix $H$ is invertible, what is refered to as LiSSA by Koh and Liang (2017) is merely the use of $H\^{-1} = \sum\_{t=0}\^{\infty} (I - H)\^{t}$, an equality that holds if and only if $\\|I - H\\|\_{2} < 1$. However, the version of LiSSA implemented in this submission, following (Koh & Liang, 2017), does not account for this condition, leading to its divergence, as shown in Figure 1.
>
>
> **Isn't it unfair to say that a method does not work while it is not correctly implemented**? One correct implementation for LiSSA is: let $A = H / (\\|H\\|\_{F}+1)$, initialize $H\^{-1}\_{0} = I$ and proceed with $H\^{-1}_{t+1} = I + (I-A)H\^{-1}\_{t}$. Then, $H\^{-1}\_t / (\\|H\\|\_{F}+1)$ converges to $H\^{-1}$. In other words, $H$ should be scaled so that LiSSA converges, and we can retrieve the result at any time by re-scaling.
>
>
> **Besides, the purpose of the convergence test is confusing**. In theory, the Schulz’s method and LiSSA are guaranteed to converge, and datainf is clearly not designed to converge. So, why bother to compare a converging method, such as Schulz’s method, to a non-converging method, like datainf? why not implement LiSSA correctly and include other converging methods, such as conjugate gradient that has also been mentioned by Koh and Liang (2017)？Whether or not a method converges can already be determined in theory.
>
> ---
> Agarwal, N., Bullins, B., & Hazan, E. (2017). Second-order stochastic optimization for machine learning in linear time. Journal of Machine Learning Research, 18(116), 1-40.
>
> Koh, P. W., & Liang, P. (2017). Understanding black-box predictions via influence functions. In International conference on machine learning (pp. 1885-1894). PMLR.

---

> > ### Author Response · Authors · 2024-12-02
> > **Follow-up of convergence test**
> >
> > Dear Reviewer mprn:
> >
> > Thanks a lot for the in-depth discussion of the implementation of LiSSA! We reimplement the algorithm following the instructions in the anonymous codebase (https://anonymous.4open.science/r/HyperINF-B702/rebuttal/converge_test.py). The results are presented in (https://anonymous.4open.science/r/HyperINF-B702/rebuttal/hvp_converge_correct_lissa.pdf), where HyperINF still shows consistently better convergence performance than LiSSA .
> >
> > Since tomorrow is the last day of the author-reviewer discussion phase, we hope our response can resolve your concerns. Let us know if you have other questions!
> >
> > Sincerely,
> >
> > the authors

---

> > > ### Comment · Reviewer_mprn · 2024-12-03
> > >
> > > Thank you for your update. I notice that it is close to the end of discussion, but I would recommend that the authors also include conjugate gradient for comparison. The convergence curve of the HyperINF makes me suspect that the level set of $f(x)=x\^{\top}Mx$, where $M$ is a matrix used for convergence test in the paper, is close to some significantly degraded ellipsoid. In this case, conjugate gradient might present similar convergence curve.
> > >
> > > Anyway, there are many converging iterative algorithms to approximate $M\^{-1}v$ in the community of optimization. I think it would be more meaningful to demonstrate that better approximation would improve downstream performance for influence function.

---

> > > > ### Author Response · Authors · 2024-12-03
> > > >
> > > > We thank the reviewer for the suggestion! We refer the reviewer to the **Appendix G**, where we include the comparison to other matrix inversion methods in **Table 9, Table 10 and Figure 11**. Comparing to Conjugate gradient (CG), Schulz's method shows better accuracy and efficiency.

---

> ### Author Response · Authors · 2024-12-04
> **End of the discussion phase**
>
> Dear Reviewer mprn,
>
> Thank you again for your efforts reviewing our paper and your engagement in discussions! Since the rebuttal phase comes to the end, we hope our rebuttals and additional results have resolved your concerns. We will appreciate it if you can consider to adjust your score if our responses are satisfying. Thanks!
>
> Sincerely,
>
> the authors

---

### Official Review · Reviewer_h5zQ · 2024-11-05

**Soundness:** 1
**Presentation:** 3
**Contribution:** 1
**Rating:** 3
**Confidence:** 5

**Summary:**

This paper proposes a new method for efficiently approximating influence function, which contains two steps: (1) based on certain assumptions, decompose the expected gradient outerproducts into the kronecker products between identity matrix and average covariance matrix of gradient columns. (2) use Schulz's method to invert the matrix.

**Strengths:**

Trending topic, the line of data attribution is very important, especially in the era of foundation models.

**Weaknesses:**

I am mainly concerned about the assumption made in Lemma 1. I seem to not find any justification for the assumption of zero expectation and independence for gradient columns where the randomness is taken over the label $y ~ p(y|}x, \theta)$. However, this seems to be the key result for the paper (I don't think the application of using Schulz's method for matrix inverse is very impressive). I also took a look at the proof for Lemma 1 and I find it poorly written, which involves typos like 'Var(g(:,k), g(:,k))' and other stuff. The only justification for the assumption I found in the paper is in line 223-224, but it's still very unclear. What does it even mean by "each column is independent and identical"? With respect to which probability distribution?

I took a look at the pseudocode of the algorithm. It seems that to compute generalized FIM, the author uses the groundtruth label $y_i$ instead of sampling from $y ~ p(y|}x_i, \theta)$. This is a mistake (but understandable), and the same mistake was also made in DataInf. Bartlett’s identities are with respect to model distribution $p(x, y, \theta) = p(x)p(y|}x, \theta)$, not the groundtruth distribution! This issue has been discussed in the optimization community https://arxiv.org/abs/1905.12558. While the two quantities might be fairly close to each other for well-trained classifiers, I am not sure about the language model. Since this issue is a common mistake, it won't lower my score for the paper, but I recommend the author include a paragraph of discussion on this issue.

**Questions:**

Why not compare with K-FAC and EK-FAC approach? https://arxiv.org/abs/2308.03296 I think this is the most relevant work as K-FAC is fairly similar to Section 3.1.

---

> ### Author Response · Authors · 2024-11-28
> **Author response to reviewer h5zQ**
>
> We thank the reviewer for the feedback and suggestions. We provide further clarifications as follows:
>
> ## W1&W2: Empirical evidence justification for Lemma 3.1 and FIM approximation
> We agree with the reviewers that our assumptions made for Lemma 3.1 are idealized and hard to verify in real large language model training. Instead, we want to demonstrate that in some conditions, the generalized fisher information matrix (GFIM) and fisher information matrix (FIM) are equal in expectation, which motivates our low-rank formulation of the information function (Equ. 5).
> We also thank the reviewer for the great reference! As the reviewer suggested, we provide a detailed discussion on the limitations of GFIM and FIM approximations in Appendix I.
>
> ## Q1: Performance comparison with EK-FAC
> We have added a new comparison to EKFAC for LLM finetuning data selection. Since it only supports the influence computation on FFN layers, we only apply it to dense finetuning experiments. Specifically, we use all the gradients on Linear modules in the last layer of Llama-2-7B to compute the influence score by EKFAC algorithm. We present the results on QASC dataset as follows, where HyperINF outperforms EKFAC by a large margin. We promise to include the complete results on all datasets in the updated version.
>
> | QASC (k%) | Random | HyperINF | EKFAC |   |
> |-----------|:------:|:--------:|:-----:|:-:|
> |     5%    |  11.3  |   14.3   |  9.7  |   |
> |    20%    |  13.3  |   15.0   |  10.3 |   |
> |    40%    |  18.1  |   56.1   |  10.6 |   |
>
> We hope our answers resolve most of your concerns. If you have any further questions, don’t hesitate to contact us. Your valuable suggestions have helped us improve the manuscript!

---

> > ### Comment · Reviewer_h5zQ · 2024-11-28
> >
> > Thanks so much to the authors for the detailed response.
> >
> > I previously did not notice that Lemma 3.1 is the same as Lemma 1 in Yang et al. (2022), as pointed out by several other reviewers. This can significantly reduce the novelty of this paper as Section 3.1 now seems to me a simple adaptation. While the authors have explicitly acknowledge that the gradient column independence assumption may be violated in practice, I suggest the authors to additionally discuss under what condition it holds. Without such a discussion, it's hard to understand why the experiment should work (it's okay not to have a theoretical guarantee here which I understand is hard).

---

> > > ### Author Response · Authors · 2024-12-02
> > > **Follow-up clarification on Lemma 3.1**
> > >
> > > Dear Reviewer h5zQ:
> > >
> > > Thanks a lot for the in-depth discussion of Lemma 3.1!
> > > We acknowledge that we follow the same Lemma as proposed by Yang et al. (2022), but this is the first time to extend it to data attribution, instead of optimization problems. With the low-rank formulation of influence function, we are able to improve both the efficiency and accuracy, which brings a empirical benefit to real-world data selection tasks.
> > >
> > > We also provide the experimental results that GFIM and FIM demonstrates similar patterns when **each column are independently and identically drawn from distribution with zero expectation**. We provide the implementation in the anonymous codebase (https://anonymous.4open.science/r/HyperINF-B702/rebuttal/fim_gfim.py) and demonstrate it with a standard Gaussian distribution.
> > >
> > > Since tomorrow is the last day of the author-reviewer discussion phase, we hope our response can resolve most of your concerns. Let us know if you have other questions!
> > >
> > > Sincerely,
> > >
> > > the authors

---

> > > > ### Comment · Reviewer_h5zQ · 2024-12-03
> > > >
> > > > Thanks for authors' response and additional experiments. However, I don't think it serves as evidence for the assumption to hold as the randomness is artificial and very far from the actual experiment setting. To fix this, you can either (1) empirically compare the influence score computed with and without Lemma 3.1 on toy examples (where you can use Schulz's method for matrix inversion) or (2) discuss under what conditions the assumption holds. As an example, the assumption of using GNM as an approximation to Hessian holds when the models are trained near convergence, and the loss function has small curvature.

---

> > > > > ### Author Response · Authors · 2024-12-03
> > > > >
> > > > > For (1), we compared HyperINF with and without GFIM in Appendix D, Figure 5, where we show using GFIM will not sacrifice accuracy with better efficiency.
> > > > >
> > > > > for (2), we have clarified in the paper the idealized conditions, i.e. the zero-mean condition can be satisfied when the model is trained to converge (Line 229-Line 231). However the i.i.d. distribution of each column is hard to be satisfied. Each column in the gradient matrices corresponds to the weights attached to one neuron. Since the neurons cannot be guaranteed to be independent, the condition is hard to met in realistic cases. We discussed other conditions in Appendix I.

---

> ### Author Response · Authors · 2024-12-04
> **End of the discussion period**
>
> Dear reviewer h5zQ,
>
> Thank you again for your efforts reviewing our paper and your engagement in discussions! Since the rebuttal phase is towards the end, we hope our rebuttals and additional results have resolved your concerns. We will appreciate it if you can consider to adjust your score accordingly. Thanks!
>
> Sincerely,
>
> the authors

---

### Official Review · Reviewer_NbyJ · 2024-11-12

**Soundness:** 4
**Presentation:** 4
**Contribution:** 3
**Rating:** 5
**Confidence:** 4

**Summary:**

This paper proposes to address the efficient and effective data attribution problem, with application to LLM and LVM fine-tuning.

The authors first based on LISSA and DATAINF, propose to attribute validation loss to individual fine-tuning examples, via a product of gradient, Hessian inverse, and a vector. The bottleneck of the computation is in the inverse of the large Hessian matrix, which has dimensionality d\times d. LISSA use the iterative method to find $H^{-1}v$, while DATAINF uses the Sherman-Morrison formula and the Fisher Information Matrix (FIM) approximating the Hessian for efficient computation of $H^{-1}v$.
The proposed HyerPower method is in Eq. (5), where the GFIM is used in place of FIM.

Schulz’s method is a hyperpower iterative method, since "Schulz’s method demonstrates superior accuracy in terms of error rate and significant efficiency gains from the GPU acceleration on matrix multiplications".

Overall, the paper used a new numerical computation method (Schulz’s method) to gain efficiency, under the framework set by previous work (influence function by Koh & Liang, 2020, and DATAINF/LISSA).

The experimental results are somehow strong:
1) simulation showed that LISSA and DATAINF won't converge, while Schulz’s method converges quickly.
2) on a LORA-tuned LLM Roberta-large, they showed that the mislabeled data points can be detected by HYPERINF more accurately.
3) on a LORA-tuned LLM (Llama2-7B), they showed that data selected by HYPERINF leads to better fine-tuning accuracy.
4) on a LVM, they showed that data selected by HYPERINF leads to better pre-training.

**Strengths:**

+ The computational time and memory usage for data attribution are greatly reduced by HYPERINF.
+ the effectiveness and efficiency are demonstrated on several tasks, showing the generality of HYPERINF.

**Weaknesses:**

- The novelty may be limited, in the sense that an existing numerical power method is applied to an existing problem (The identification of the challenge and the solution are still recognized).
- There are some experimental observations that are not explained well. See the questions

**Questions:**

In Table 2 & 4, it shows that when selecting a smaller portion of bad data, the improvement over DATAINF is very limited (e.g., in Table 4, HYPERINF has 53.2 and the runner-up has 53). Any explanation about this?

In Table 3, why dense fine-tuning gives worse performance than sparse fine-tuning?

Did you try several initialization for LISSA and DataInf? HYPERINF may have unstability issue too and multiple initializations should be tried.

Can you make your contributions in Eqs. (4-6) clear by comparing your methods to existing technique? It seems that these equations are adopted from previous work.

---

> ### Author Response · Authors · 2024-11-28
> **Author response to reviewer Nbyj**
>
> We thank the reviewer for the thoughtful feedback and valuable suggestions! We provide the clarifications as follows:
>
> ## W1&Q4: Clarification on the main contributions
> The first contribution of this work is leveraging Generalized Fisher Information (GFIM, Equ. 4) in the data attribution problem, which yields a novel low-rank formulation of the data influence estimation function (Equ. 5). Secondly, we apply Schulz’s method (Equ. 6) to improve the accuracy of matrix inverse approximation. The mathematical formulas of Schulz’s method are from very old linear algebra literature, while it has not been used in large-scale neural network training or data attribution problems. By combining the two techniques, we can achieve improved efficiency and accuracy, which then transfer to better empirical performance. In comparison, the prior work (DataInf, [1]) only proposed to apply the Sherman-Morrison formula to the previously proposed influence function. We consider our contribution is not incremental compared to DataInf.
>
> ## Q1: Limited improvement over DataInf
> We agree with the reviewer that HyperINF could perform comparably as DataINF with LORA finetuning on datasets with little noise. However, it shows a large improvement over DataINF with dense finetuning (Table 3). We provide a potential explanation based on the error analysis: The approximation error of DataInf algorithms is $O(d^2)$ according to [1]. With LoRA fine-tuning, the number of tunable parameters per layer is relatively small, whereas DataInf could yield a small approximation error which has less impact on the rank of data samples. In the dense finetuning case (Table 3), HyperINF could have a larger improvement above DataInf when we have a larger amount of tunable parameters.
>
> ## Q2: Worse performance with dense finetuning than low-rank finetuning
> According to the original LoRA paper (Table 2 in [2]), using LoRA finetuning could possibly lead to better performance than dense finetuning (FT).
>
> ## Q3: Different distributions of matrix used in convergence test
> We agree with the reviewers that difference algorithms could show various convergence performances on matrices drawn from different distributions. We have updated the paper to include more holistic convergence test results in **Appendix H** with five different matrix distributions. We also include the results from Neumann series and Successive Over Relaxation (SOR), while HyperINF consistently converges to a desirable error, while other methods could be sensitive to the distributions and scale of the targeted matrices.
>
> We hope our answers resolve most of your concerns. If you have any further questions, don’t hesitate to contact us. Your valuable suggestions have helped us improve the manuscript!
>
> [1] DataInf: Efficiently Estimating Data Influence in LoRA-tuned LLMs and Diffusion Models
>
> [2] LoRA: Low-Rank Adaptation of Large Language Models

---

> ### Author Response · Authors · 2024-12-02
> **End of the discussion period**
>
> Dear reviewer NbyJ,
>
> Thank you again for your efforts reviewing our paper and providing valuable feedback! Since **tomorrow is the last day of the author-reviewer discussion phase**, we hope you can take some time to look into our rebuttals and additional results. Let us know if you have other concerns and questions!
>
> Sincerely,
>
> the authors

---

### Official Review · Reviewer_XXD3 · 2024-11-13

**Soundness:** 3
**Presentation:** 3
**Contribution:** 2
**Rating:** 6
**Confidence:** 3

**Summary:**

This paper identifies the lack of convergence in computing hessian inverse to be the primary source of poor performance of existing data attributions methods. To alleviate this, the authors look to the family of hyperpower methods. Specifically, the authors leverage the Schulz method for its convergence guarantees. They show promising empirical performance (with minimal computational overhead) on various data attribution / mislabelling tasks.

**Strengths:**

- The problem that this paper addresses is a challenging one, and one of increasing importance/popularity in the community.
- The writing is pretty clear, and the experiments are well described.
- The proposed method is sound, and can potentially see adoption in the community/real world.

**Weaknesses:**

- The main contributions of this paper are not clearly disentangled from the overall story. In particular, my understanding is that the primary contribution of this paper is identifying that the Schulz method from the matrix inverse can be efficiently applied in this setting. The rest of the pipeline (Hessian inverse based attribution, Fisher Information Matrix etc) is borrowed from existing work in the field.
- I'm hesitant to use the term marginal/ limited novelty as the authors have made an important observation about an important problem and proposed a (albeit, given previous work, straightforward) method to alleviate it. However, the way the paper is structured currently certainly makes the paper seem like it has very limited novelty.

**Questions:**

- What do you view as the single main contribution of this paper?
- If the main contribution is indeed the incorporation of the Schulz method, was there a reason the paper was positioned and written as a "new data attribution method" instead of "evaluating better matrix inverse methods for data attribution"?
- I notice you have included some discussion of other matrix inverse methods in Appendix F. How were the matrices chosen for the experiments? Were they random, or drawn from some real world data attribution settings? Are certain methods better than others for certain distributions of matrices?
- Did you try other techniques for matrix inversion such as Neumann series approximation and successive over relaxation?

---

> ### Author Response · Authors · 2024-11-28
> **Author response to reviewer XXD3**
>
> ## W1&W2&Q1&Q2: Clarification on the main contributions
> We thank the reviewer for recognizing the importance of our target problem! The main problem we want to solve is the hessian-inverse approximation with satisfying accuracy and efficiency. We did follow the influence function formulation for data attribution while incorporating Generalized Fisher Information instead of FIM in previous works.
>
> The first contribution of this work is leveraging Generalized Fisher Information (GFIM, Equ. 4) in the data attribution problem, which yields a novel low-rank formulation of the data influence estimation function (Equ. 5). Secondly, we apply Schulz’s method (Equ. 6) to improve the accuracy of matrix inverse approximation. The mathematical formulas of Schulz’s method are from very old linear algebra literature, while it has not been used in large-scale neural network training or data attribution problems. By combining the two techniques, we can achieve improved efficiency and accuracy, which then transfer to better empirical performance. In comparison, the prior work (DataInf, [1]) only proposed to apply Sherman-Morrison formula to the previously proposed influence functions. We consider our contribution is not incremental compared to DataInf. **We have updated the paper to include a list of contribution in Line 81 - Line 93.**
>
> ## Q3: Ablation on the distribution of matrices used in convergence test
> ## &Q4: Comparisons to other matrix inversion techniques
> We agree with the reviewers that difference algorithms could show various convergence performances on matrices drawn from different distributions. We also have updated the paper to include more holistic convergence test results in **Appendix H** with five different matrix distributions. We also include the results from Neumann series and Successive Over Relaxation (SOR), while HyperINF consistently converges to a desirable error, while other methods could be sensitive to the data distributions and the scale of the targeted matrices.
>
> [1] DataInf: Efficiently Estimating Data Influence in LoRA-tuned LLMs and Diffusion Models

---

> > ### Comment · Reviewer_XXD3 · 2024-11-30
> >
> > Thanks for the response.
> >
> > While I still maintain that the techniques aren't particularly novel, I see some practical value in identifying and showing the efficiency of this useful (albeit simple) combination of ideas to the community.
> >
> > However, the comment of `reviewer mprn` about unfair comparison to lissa has certainly caught my attention as it brings to question to true in-practice effect of incorporating the Schulz method. Thus, I'm willing to raise my score to 6 contingent on a satisfactory resolution of the same.

---

> > > ### Author Response · Authors · 2024-12-02
> > >
> > > We thank the reviewer for adjusting the score! We have reimplemented LiSSA according to reviewer mrpn's suggestion and present the results in the anonymous codebase (https://anonymous.4open.science/r/HyperINF-B702/rebuttal/hvp_converge_correct_lissa.pdf). The result demonstrates that HyperINF still consistently outperform LiSSA after the correction.
> > >
> > > We also thank the reviewer for the in-depth discussion on our implementation of GFIM and Schulz's algorithm!
> > > We acknowledge that we follow the same Lemma as proposed by Yang et al. (2022), but this is the first time to extend it to data attribution, instead of optimization problems. With the low-rank formulation of influence function, we are able to greatly improve both the efficiency and accuracy, which brings a empirical benefit to real-world data selection tasks.
> > > We are also willing to run more experiments in various benchmarks and settings (e.g. with different level of noises) and enrich our final manuscript.
> > >
> > > Sincerely,
> > >
> > > the author

---

### Author Response · Authors · 2024-11-28
**General Response**

We thank all the reviewers for their thoughtful and valuable feedback, which helps us improve our current work! We summarize the main concerns from reviewers and give our corresponding clarifications as follows:

1. **Q: What is the main contribution of the work?** [XXD3, Nbyj, mprn, v6An, 3Qw4]

    A: The first contribution of this work is leveraging Generalized Fisher Information (GFIM, Equ. 4) in the data attribution problem, which yields a novel low-rank formulation of the data influence estimation function (Equ. 5). Secondly, we apply Schulz’s method (Equ. 6) to improve the accuracy of matrix inverse approximation. The mathematical formulas of Schulz’s method are from very old linear algebra literature, while it has not been used in large-scale neural network training or data attribution problems.

2. **Q: Have you tried matrices from different distributions on the convergence test?** [XXD3, Nbyj, mprn, GxRJ]

    A: We agree with the reviewers that difference algorithms could show various convergence performances on matrices drawn from different distributions. We have updated the paper to include more holistic convergence test results in Appendix H with five different matrices distributions. We also include the results from Neumann series and Successive Over Relaxation (SOR), while HyperINF consistently converges to a desirable error, while other methods could be sensitive to the distributions and scale of the targeted matrices.

3. **Q: Why are the time costs of HyperINF and DataINF independent from the LORA rank? Lack of a detailed discussion on complexity and computation overheads from algorithms.** [GxRJ, 3Qw4]

    A: We thank the reviewer for this intriguing observation! We have also discovered this discrepancy between theoretical complexity and real GPU hours, and we are happy that we may have found the answer. According to **Appendix D.1**, we provide both the time costs on CPU and GPU computing across various LoRA ranks. We observe that the efficiency of three algorithms ranks largely differently between GPU and CPU. On CPU (Figure 6), DataInf introduces the least time overheads while HyperINF incurs the most amount of extra time costs. In addition, the time costs from DataInf and LiSSA increase quadratically with LoRA rank $r$ while HyperINF increases linearly. Alternatively, on one single GPU (Figure 7), the time costs from all algorithms are almost constant across LORA ranks, and HyperINF costs the least amount of time, followed by DataInf. In comparison, LISSA requires ~4x more time costs than HyperINF and DataInf. That difference demonstrates that HyperINF is more efficient and compatible with modern GPU computing.

4. **Q: Can you compare to more baseline methods on data attribution (e.g. EKFAC)?** [h5zQ]

    A: We have added a new comparison to EKFAC for LLM finetuning data selection. Since it only supports the influence computation on linear layers, we only apply it to dense finetuning experiments. Specifically, we use all the gradients on Linear modules in the last layer of Llama-2-7B to compute the influence score by EKFAC algorithm. We present the results on the QASC dataset as follows, where HyperINF outperforms EKFAC by a large margin. We promise to include the complete results on all datasets in the updated version.

| QASC (k%) | Random | HyperINF | EKFAC |   |
|-----------|:------:|:--------:|:-----:|:-:|
|     5%    |  11.3  |   14.3   |  9.7  |   |
|    20%    |  13.3  |   15.0   |  10.3 |   |
|    40%    |  18.1  |   56.1   |  10.6 |   |

5. **Q: Can you provide more clarification and empirical justification on your assumption for Lemma 3.1, where you assume each column of the gradient matrix is independent and identical?** [h5zQ,mprn,GxRJ,7qzk]

    A: We agree with the reviewers that our assumptions made for Lemma 3.1 are idealized and we do not claim they hold for real large language model training. Instead, we want to demonstrate that under some conditions, the generalized fisher information matrix (GFIM) and the Fisher information matrix (FIM) agree in expectation, which motivates our low-rank formulation of the information function (Equ. 5). As reviewers suggested, we now also included a detailed discussion on the limitations of GFIM and FIM approximations in real LLM training cases in **Appendix I**.

For other questions and points of clarification, we provide detailed responses correspondingly under each individual review.

---

### Meta-Review · Area_Chair_9NHA · 2024-12-20

**Metareview:**

this paper had several interesting discussions between the reviewers and the authors. The reviewers are appreciative of the paper's writing and evaluations, but the general consensus is that the paper falls short of the acceptance bar, mostbly becase of lack of novelty in theoretical contributions with assumptions that are too idealized to have any real world relevance. The paper could also improve more from deeper empirical evaluations and discussions.

**Additional Comments On Reviewer Discussion:**

The reviewers are in consensus of lack of novelty of theoretical contributions. Some reviewers asked for more experimenets which the authors obliged with. However there are still some concerns remaining e.g. about a random selection performing better than many existing methods.

---

### Decision · Program_Chairs · 2025-01-22

Reject